# Tau propagation in the brain olfactory circuits is associated with smell perception changes in aging

Ibai Diez [1,2,11], Laura Ortiz-Terán [1,3,11], Thomas S. C. Ng [4,5], Mark W. Albers [6], Gad Marshall[7], William Orwig [1,8], Chan-mi Kim[1], Elisenda Bueichekú [1], Victor Montal[1,9], Jonas Olofsson [10], Patrizia Vannini [2,7], Georges El Fahkri[1], Reisa Sperling [2,6,7], Keith Johnson[1], Heidi I. L. Jacobs [1] & Jorge Sepulcre [1,2] ✉

The direct access of olfactory afferents to memory-related cortical systems has inspired theories about the role of the olfactory pathways in the development of cortical neurodegeneration in Alzheimer's disease (AD). In this study, we used baseline olfactory identification measures with longitudinal flortaucipir and PiB PET, diffusion MRI of 89 cognitively normal older adults (73.82 ± 8.44 years; 56% females), and a transcriptomic data atlas to investigate the spatio-temporal spreading and genetic vulnerabilities of AD-related pathology aggregates in the olfactory system. We find that odor identification deficits are predominantly associated with tau accumulation in key areas of the olfactory pathway, with a particularly strong predictive power for longitudinal tau progression. We observe that tau spreads from the medial temporal lobe structures toward the olfactory system, not the reverse. Moreover, we observed a genetic background of odor perception-related genes that might confer vulnerability to tau accumulation along the olfactory system.

Olfaction is a sensory modality with distinctive connectomic features compared to other senses, such as vision, audition, or touch. The afferences from the olfactory system – olfactory epithelium, olfactory nerve, and olfactory bulb – bypass the thalamic relay stations to connect the cortex directly to the ipsilateral piriform cortex[1–3]. While other major sensory systems also interact with the limbic system and brainstem regions, the olfactory tract directly projects to the limbic system – structures in the medial temporal lobe such as the amygdala and entorhinal cortex – as well as the brainstem, particularly to the locus coeruleus[4]. This prompt and direct access from peripheral afferences to memory and cognitive-related cortical systems in the human brain has inspired theories about the potential role of the olfactory systems as a gateway for infectious or environmental agents that might trigger cortical neurodegeneration[5]. The early affectation of olfaction functionality not only in aging[6,7] but also in Alzheimer's disease (AD), Parkinson's disease, Huntington's disease, and amyotrophic lateral sclerosis, amongst others[8], has prompted speculation in this regard. For instance, olfactory deficits are present in 90% of patients with AD[9] and emerge in preclinical stages, many years before cognitive decline and clinical diagnosis[10], potentially because pathological

[1]Gordon Center for Medical Imaging, Department of Radiology, Massachusetts General Hospital, Harvard Medical School, Boston, MA, USA. [2]Athinoula A. Martinos Center for Biomedical Imaging, Department of Radiology, Massachusetts General Hospital, Harvard Medical School, Charlestown, MA, USA. [3]UMASS Memorial Medical Center, UMASS Chan Medical School, Worcester, MA, USA. [4]Division of Nuclear Medicine and Molecular Imaging, Department of Radiology, Massachusetts General Hospital, Harvard Medical School, Boston, MA, USA. [5]Center for Systems Biology, Massachusetts General Hospital, Harvard Medical School, Boston, MA, USA. [6]Department of Neurology, Massachusetts General Hospital, Harvard Medical School, Charlestown, MA, USA. [7]Department of Neurology, Brigham and Women's Hospital, Harvard Medical School, Boston, MA, USA. [8]Harvard University, Department of Psychology, Cambridge, MA, USA. [9]Barcelona Supercomputing Center, Barcelona, Spain. [10]Stockholm University, Department of Psychology, Stockholm, Sweden. [11]These authors contributed equally: Ibai Diez, Laura Ortiz-Terán. ✉e-mail: jorge.sepulcre@yale.edu

aggregates have been found at early stages in the olfactory structures[11–15]. How these aggregates reach the olfactory system is not clear. While some theories propose that aggregates reach the olfactory system from brainstem subnuclei, similarly as they might emerge from the brainstem to the medial temporal cortex, there are also hypotheses suggesting a link with viral infections[5,16] and other xenobiotic agents[17,18] on the olfactory pathways, in which the AD-related pathology builds up first within the olfactory system periphery and later spread to the limbic cortex.

AD is characterized by an extra-neuronal accumulation of amyloid-$\beta$ and an intraneuronal deposits of neurofibrillary tangles[19]. Tau pathology has been demonstrated to be the best indicator for AD progression compared to other imaging modalities. Braak and Del Tredici[20] suggested that tau pathology may initially develop in the locus coeruleus during young adulthood[21] and advance later in life to trans-entorhinal regions (Stage I) and subsequently to the entorhinal, hippocampal formation, and the olfactory bulb[22] (Stage II). This is followed by propagation to the basal temporal cortex (Stage III) towards the temporal, insular, and frontal neocortex (Stage IV). It finally reaches neocortical association areas (Stage V) and primary regions (Stage VI). While olfactory assessment has been proposed as a biomarker for preclinical AD diagnosis[23–26] and its dysfunction has been repeatedly correlated with cognitive decline in healthy aging[27] and grey matter volume loss[28,29], its association with cortical tau spreading is still unknown. Characterizing tau spreading across the olfactory system, medial temporal lobe, and brainstem is vital to better understanding AD-related early pathological hallmarks. How tau aggregation accumulates over time in the olfactory system compared to other systems represents an opportunity to elucidate crucial connectomic and biological mechanisms about the early stages of AD and the potential relationship between internal and external risk factors entering the human brain via the olfactory system.

In this study, we designed a multimodal strategy to integrate connectomic, olfaction, and neuroimaging-genetic data to investigate the in-vivo spreading patterns implicated in the tau accumulation of the olfactory cortical system and identify olfactory test biomarkers that maximize the prediction of tau accumulation and longitudinal spreading. We hypothesize that olfactory dysfunction will result from tau accumulation along olfactory-related pathways from brainstem and medial temporal regions accelerated by amyloid accumulation, prior to cognitive symptoms appearing, and that olfactory genes will show brain spatial expression patterns resembling tau progression. We used cross-sectional and longitudinal olfaction performance via olfactory identification tests, flortaucipir and PiB positron-emission tomography (PET), structural and diffusion tensor imaging of the Harvard Aging Brain Study (HABS)[30] cohort – an ongoing longitudinal observational study designed to study the differentiation of AD from healthy aging. This overall strategy allowed us to characterize key indicators of tau spreading over time related to olfaction impairment and provide support asserting the predominance of an intrinsic vulnerability of the olfactory systems to develop tau pathology rather than an axis of pathological spread from peripheral to central regions, as would be expected if there was an infectious or environmental external cause.

## Results

This study included 418 clinically normal HABS participants with baseline smell identification measurement. Of these participants, 155 underwent a flortaucipir PET tau acquisition within six months of the smell test and 89 a follow-up scan ~2.5 years after the baseline. All included participants had a 0 score on the Clinical Dementia Rating scale, an MMSE score $\geq 25$, and performed within education-adjusted norms on the logical memory delayed recall test (see Methods section).

### Odor Identification, Neurodegeneration, and Memory

We found a curvilinear relationship between odor identification measurement and age ($N = 418$; Fig. 1a), observing a particular decay beyond the seventh decade of life. The distance of each participant to the quadratic function identified several individuals strongly deviating from the norm (decreased odor identification abilities below the corresponding age group). The 40 different odorants used in the UPSIT composite test were further studied to identify the percentage of participants correctly identifying each odorant (Fig. 1b). While some odorants such as onion, peanut, and smoke were correctly identified by most of the participants (>90% identification rate), others were more difficult to identify, such as lime and lemon (<60% identification rate).

Next, we evaluated the number of odorants that by themselves were associated with atrophy or tau accumulation at voxel-level ($N = 155$; using general linear models controlling for age, sex, APOE$\epsilon$4 status, and smoking history; Fig. 1c and d). Tau accumulations in medial temporal regions, limbic areas, temporo-parietal regions, and dorsolateral prefrontal cortex were associated with poor identification of at least seven different odorants (Fig. 1c). Interestingly, tau in olfactory regions (piriform cortex, anterior olfactory nucleus, entorhinal, and amygdala) was associated with higher incidence of incorrectly identified odor items (insets in Fig. 1c). An additional analysis focused on grey matter atrophy (voxel-based morphometry) showed congruent results. Medial temporal structures (entorhinal, parahippocampus, anterior insula) and primary olfactory regions were associated with the highest number of incorrectly identified odor items.

We also investigated whether there was a relationship between the odor identification composite and cognitive score (PACC composite), controlling for age, gender, APOE$\epsilon$4 status, smoking history, and years of education ($N = 155$; Fig. 1e). We found a significant association between baseline odor identification and cognitive score ($r = 0.20$; T-statistic = 2.45; $p = 0.015$), which was stronger when performing associations between odor identification at baseline and cognitive performance ~2.5 years later ($r = 0.31$; T-statistic = 3.92; $p = 0.0001$; Fig. 1e). This association was still significant when controlling for PACC at baseline ($r = 0.24$; T-statistic = 2.95; $p = 0.0004$).

### Brain Mapping of Olfaction Performance in Cross-sectional and Longitudinal Tau Imaging

We used regression analysis to investigate the voxel-level association between cortical tau accumulation – measured at baseline and after ~2.5 years of follow-up – and the UPSIT smell test (composite of the 40 odorants) in 89 participants with longitudinal data, while controlling for age, sex, APOE$\epsilon$4 status, and smoking history (Fig. 2). At baseline, we observed worst smell identification performance related to greater tau binding in primary olfactory regions, including the piriform cortex, anterior olfactory nucleus, amygdala, and entorhinal (volume insets in Fig. 2a). Regions associated with the 3rd Braak Stage were also identified unilaterally in the right hemisphere, including the temporal pole, inferior temporal areas, anterior insula, and orbitofrontal regions (cortical maps in Fig. 2a; results adjusting for age, sex, and smoking history in Supplementary Fig. 1). The results remained congruent when adjusting the regression analysis for the presence of atrophy or amyloid (Supplementary Fig. 2). Next, longitudinal data were used to study whether the baseline UPSIT smell test could predict tau progression over time (Fig. 2b). Tau accumulation showed significant associations with the odor task scores in the primary olfactory regions (volume insets in Fig. 2b), as well as the medial prefrontal cortex, anterior cingulate cortex, and parietal regions, with an extended representation of olfactory-related areas (cortical maps in Fig. 2b, see Supplementary Table 5 for cross-sectional and longitudinal associations between UPSIT and individual medial temporal and olfactory regions).

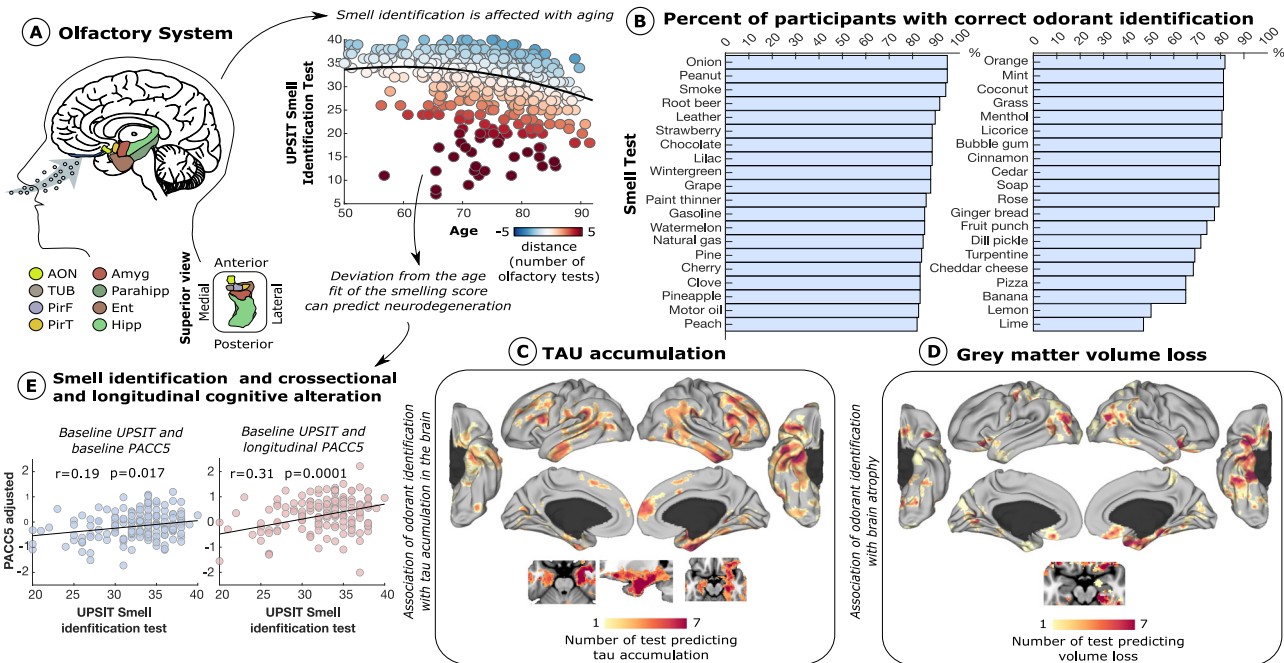

**Fig. 1 | Olfactory system dysfunction in healthy aging and its association with neurodegeneration. A** Primary olfactory regions of the brain and smell identification ability in healthy aging are displayed. The colors in the scatterplot represent the deviation of each participant's estimated smell identification test based on their age ($N = 418$). **B** Sorted bar plot of the percentage of participants correctly identifying the 40 odorants composing the UPSIT test. **C, D** Amount of odorants able to distinguish high accumulation of tau and atrophy at voxel-level ($N = 155$).

**E** Association between baseline smell identification and cognitive performance at baseline and after ~ 2.5 years using a linear regression model adjusting for age, gender, APOEε4 status, smoking history, and years of education ($N = 155$). AON anterior olfactory nucleus, TUR olfactory tubercle, PirF frontal piriform cortex, PirT temporal piriform cortex, Ent entorhinal, Amyg amygdala, and Hipp hippocampus. Source data are provided as a Source Data file.

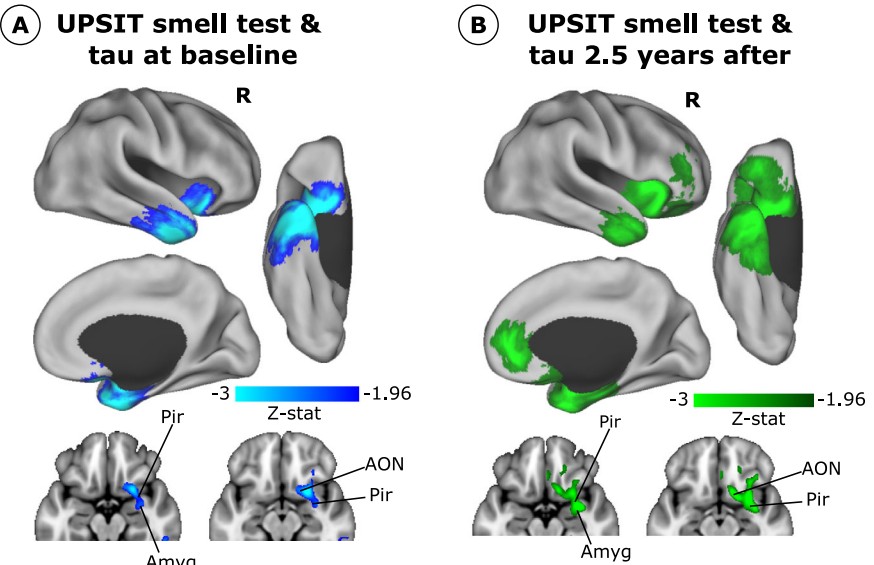

**Fig. 2 | Smell identification test association with tau accumulation and spreading ($N = 89$). A** Z-stat of the association between UPSIT smell identification test with baseline tau deposition. **B** Z-stat of the association between baseline UPSIT smell identification test and tau accumulation ~2.5 years after. A voxel-wise

general linear model was used to compute these associations. Only results surviving cluster-wise multiple comparisons are displayed with a *p*-value < 0.05. AON anterior olfactory nucleus, Pir piriform cortex, Amyg amygdala.

## Olfactory biomarkers predicting tau accumulation and longitudinal increase

Next, we repeated the same analysis for each of the 40 odor items in the UPSIT test and computed the associations with tau and amyloid accumulation – controlling for age, sex, APOEε4 status, and smoking history ($N = 155$). We generated a network of 40 odorants x (tau

voxels + amyloid voxels) with the Z-statistic of the association between each pair. Principal component analysis was used to check the linear combination of odor items, maximizing the prediction of tau accumulation (Fig. 3). Cherry, leather, peach, and cinnamon were the odorants with the most strongly weighted components, maximizing the explained variance of the prediction of tau and amyloid

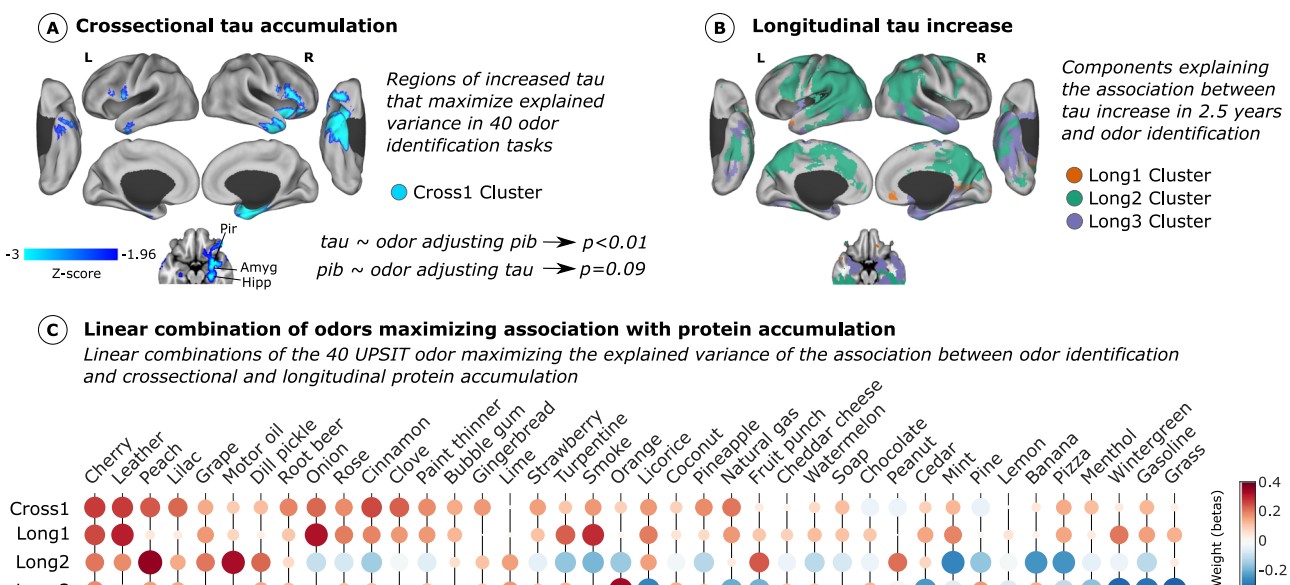

**Fig. 3 | Olfactory biomarkers of cross-sectional and longitudinal tau accumulation.** We performed a dimensionality reduction approach (principal component analysis; PCA) on a bipartite network of the association between odor item identification and tau and amyloid accumulation. **A** We obtained a linear combination of odorants that maximized cross-sectional tau accumulation prediction. Olfactory and medial temporal regions are strongly associated with odor identification information (Cross 1 Cluster; $N = 155$). **B** The same approach was repeated with the longitudinal tau data (~2.5 years), and three different clusters with different odor combinations were found representing tau accumulation at different brain regions - disease stages (Long1, Long2, Long3 clusters; $N = 89$). A voxel-wise general linear model was used to compute the association of tau accumulation with the linear combination of odorants **A** and **B**. Only regions surviving multiple comparisons with a $p$-value < 0.05 are shown. All the previous associations were adjusted for age, sex, APOEε4 status, and smoking history. Results survived when adjusting for amyloid. See Supplementary Fig. 3 for association with amyloid. Tau accumulation was associated with odor identification when controlling for amyloid but not vice versa. **C** Displays the linear combination of 40 odor items (olfactory biomarkers) predicting cross-sectional and longitudinal tau accumulation in the (**A, B**) clusters. Both the size of the circle and the color represent the coefficients of PCA, the contribution of each variable to the component (the importance of the odorant to predict tau accumulation). Source data are provided as a Source Data file.

accumulation in the brain with 23% of the brain protein accumulation variance. This component was significantly associated with higher tau accumulation in previously observed regions – including right olfactory areas, hippocampus, amygdala, temporal pole, anterior insula, inferior frontal gyrus, and orbitofrontal region (cortical map in Fig. 3a). We also observed that this component was associated with widespread amyloid accumulation throughout the brain (cortical map in Supplementary Fig. 3). The linear combination of the odorants was associated with both tau accumulation ($N = 155$; T-statistic = 3.62; $p = 0.0004$) and amyloid ($N = 155$; T-statistic = 2.83; $p = 0.005$) controlling for age, sex, APOEε4 status, and smoking history. When additionally controlling for the other PET modality, the odor composite showed association only with tau adjusting for amyloid ($N = 155$; T-statistic = 2.80; $p = 0.006$), but not for amyloid adjusting for tau ($N = 155$; T-statistic = 1.7; $p = 0.09$). We repeated the analysis with longitudinal tau data, ~2.5 years of follow-up ($N = 89$; Fig. 3b). Three different components explained 30% of the variance for the association of odor identification with longitudinal tau increase ($N = 89$; cortical map in Fig. 3b; odorants weights in Fig. 3c). A linear combination of strong weights with leather, onion, and smoke predicted a tau increase in the medial prefrontal cortex and posterior cingulate cortex (13% of the variance). A second linear combination with weights of peach and motor oil predicts a tau accumulation in Braak 5 regions – precuneus and temporo-parietal regions (9% of the variance). Finally, orange was a strongly weighted odor in the final component, predicting a tau increase in olfactory and medial temporal regions (Braak 2 and 3; 8% of the variance). Highly correlated, but different, odorants may be lowly weighted due to collinearity when linearly combined with a strongly weighted odorant (Supplementary Fig. 4 shows the Jaccard similarity index between odor items for $N = 418$). Additionally, we identified significant individual odorants predicting tau in each Braak stage (Supplementary Fig. 5).

## Connectivity disruptions and Tau spreading toward the olfactory system

To gain insights into whether connectivity disruptions and tau accumulation might play a role in the impairment of the olfactory system, we studied the olfactory pathways of the three central systems known to display connectivity to 1) the piriform cortex, 2) the medial temporal lobe (particularly the amygdala and entorhinal cortex), and to a lesser extent, 3) the brainstem. (Fig. 4). We used probabilistic tractography to characterize the degree of white matter fibers from the olfactory tract or locus coeruleus that pass through the anterior olfactory nucleus, olfactory tubercle, piriform cortex, amygdala, and entorhinal (Fig. 4a). A general linear model was used to relate the Mean Diffusivity (MD) values (a proxy of microstructural damage) at each segment with the UPSIT smell test to investigate the relevance of white matter tracts with odor identification impairment in our sample controlling for age, sex, APOEε4 status, and smoking history ($N = 82$; Fig. 4a). We found that the MD of the olfactory tract was associated with olfactory dysfunction only when entering the piriform cortex and amygdala (left linear plot; Fig. 4a). We also observed that the MD of the locus coeruleus tract shows significant associations with odor impairment when passing through brainstem nuclei like the dorsal raphe nucleus and later in the amygdala or right entorhinal cortex (right linear plot; Fig. 4a). The cortical MD statistical analysis also showed an association of the right entorhinal cortex with odor impairment.

Next, we investigated tau propagation in the olfactory system using the anatomical parcellation scheme from the MD approach and a bipartite graph theory analysis of cross-sectional and longitudinal data. This approach measures the longitudinal connectivity of tau accumulation in the brain as a proxy for longitudinal tau spreading – the association of how longitudinal tau increases in a region is related to baseline tau measures[31]. For each pair of olfactory, medial temporal,

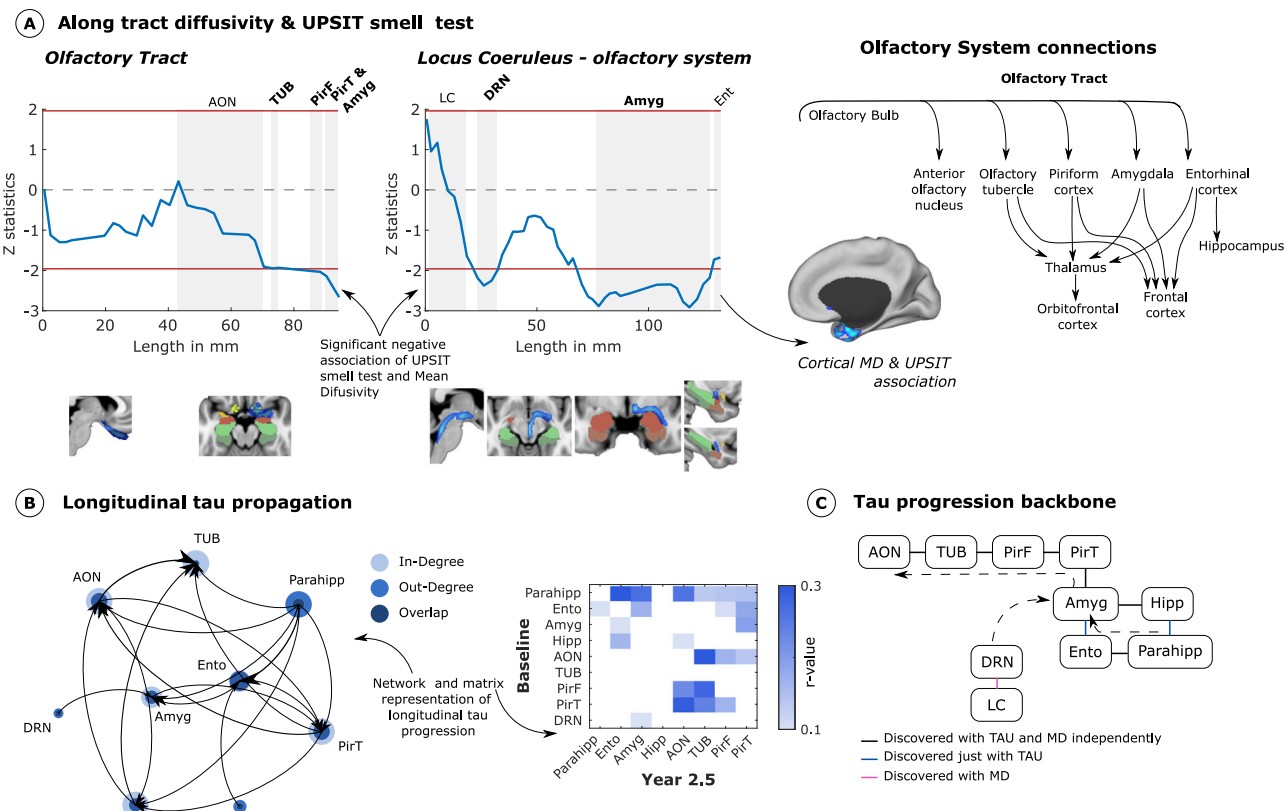

**Fig. 4 | Spreading of tau to olfactory regions. A** Along-tract statistics of the association of microstructural damage and UPSIT smell test are shown on both the olfactory tract and fibers connecting the locus coeruleus to the olfactory system ($N = 82$). A general linear model was used to compute the statistics across the tracts, adjusting for age, sex, APOEε4 status, and smoking history. Grey regions represent tract regions surrounding or passing anatomical structures of interest. Brain slices showing the evaluated tract and a diagram of known olfactory system connections are also displayed. **B** Bipartite graph between baseline and longitudinal tau accumulation showing a progression of tau from medial temporal regions towards the olfactory tract. The mean diffusivity of the dorsal raphe nucleus was also included. The node size represents the weighted degree and amount of out and in tau propagation. **C** Tau progression backbone between olfactory and medial temporal regions computed with conditional independent testing. Arrows were drawn based on directionality results obtained from a bipartite graph. AON anterior olfactory nucleus, TUR olfactory tubercle, PirF frontal piriform cortex, PirT temporal piriform cortex, Ent entorhinal, Amyg amygdala, Hipp hippocampus, Parahipp parahippocampus, LC Locus Coeruleus, DRN Dorsal Raphe Nucleus. Source data are provided as a Source Data file.

and brainstem regions, we computed the linear regression between cross-sectional tau data (as one side of the bipartite graph) and the longitudinal tau data (as the other side of the bipartite graph) adjusting for age, sex, APOEε4 status, and smoking history (Fig. 4b). We found that medial temporal regions (amygdala, parahippocampus, and entorhinal) have central roles in the longitudinal spreading of tau toward regions of the olfactory system (network graph in Fig. 4b). Moreover, we used an independence-based causal discovery algorithm (for more details, see methods section) on baseline tau to obtain the main tau propagation backbone, removing indirect connections (conditional independence testing approach; Fig. 4c). The resulting graph resembles the olfactory tract connectivity (composed of AON, TUB, Pir, and amygdala) and integrates a loop of medial temporal structures (composed of amygdala, entorhinal, parahippocampus, and hippocampus). The intersection of this graph with the mean diffusivity data shows the temporal directionality of tau progression, in which the brainstem and the medial temporal regions are key spreaders toward the olfactory system (Fig. 4c).

### Tau Spreading, Neurodegenerative Pathways, and Olfactory Perception Genes

As a final step, to investigate the relationship between tau pathology and the olfactory system in the aging brain, we used a neuroimaging-genetic approach to examine whether tau is predisposed to accumulate across the olfactory system due to in situ genetic background

(Fig. 5). We used the Allen Human Brain transcriptome dataset (AHBA) to assess the spatial distribution of gene expression levels of 390 genes related to the sensory perception of smell (GO:0007608). A co-expression matrix of the smell-related genes was generated, and four distinct co-expression clusters were identified. We used GeneCards Suite to identify the traits associated with each olfactory gene. We only kept those traits related to neurodegeneration and grouped them into six domains: tau, amyloid, Alzheimer's disease, aging, cognition, and brain morphometry (Supplementary Fig. 6 shows the associations with original traits). Twenty-four olfactory genes were associated with tau accumulation, most located in the first and fourth clusters. On the other hand, only three olfactory genes were related to amyloid accumulation. Alzheimer's and aging traits were linked with 13 genes, different genes each, mainly in the second cluster. *TOMM40* stands out in cluster 3, associated with smell perception and all the domains related to neurodegeneration. The mean spatial expression of the transcriptome of each cluster showed that their spatial distribution resembles that of tau propagation.

## Discussion

Why and how pathological aggregates settle in the olfactory system of the human brain during neurodegenerative processes, affecting its functionality, is not entirely understood. Despite investigations advocating for the use of olfactory measures as a diagnostic and prognostic biomarker for neurogenerative diseases[13,14,32,33], it remains unclear

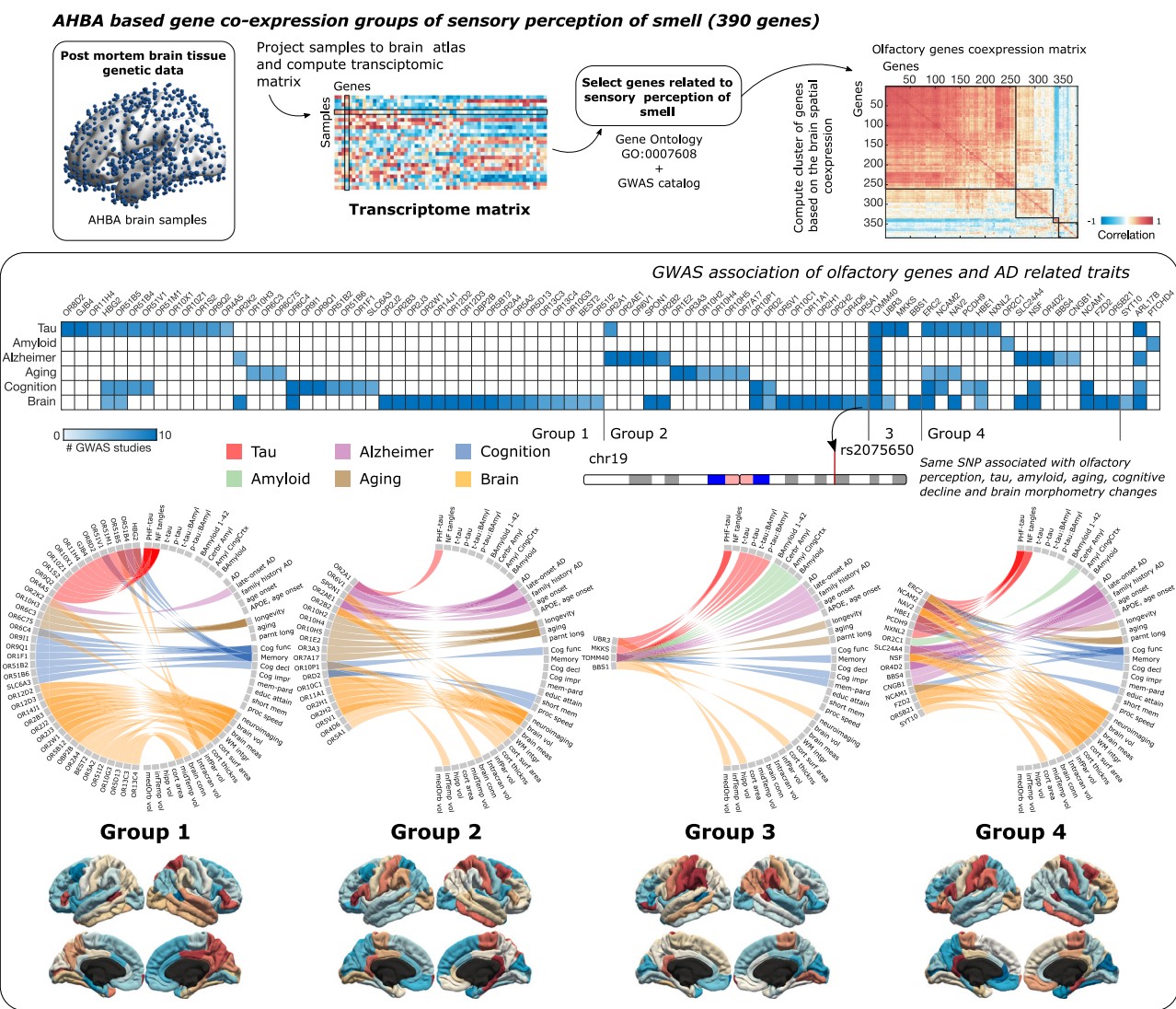

**Fig. 5 | Co-expression of genes related to olfactory perception and associated neurodegenerative traits.** Brain transcriptome data from AHBA was projected to the Desikan brain atlas and the spatial brain co-expression of 390 genes annotated as sensory perception of smell were computed. Four groups of genes with similar co-expression were identified. Using the GWAS Catalog, phenotypes associated with olfactory genes in Genecard Suite were used to calculate bipartite networks associating its genes with traits related to neurodegeneration (see Supplementary Fig. 6). For each group of co-expressed olfactory genes, a circular diagram is shown with the associations and their mean brain expression. These associations were classified into five domains: tau, amyloid, Alzheimer's disease, aging, cognition, and brain. For visualization purposes, the brain maps of the mean expression of each cluster were z-scored separately for each hemisphere. A matrix representing the association of each gene with each domain is also displayed to identify the association of each gene with the different domains. Source data are provided as a Source Data file.

whether the olfactory system is critical to understanding aging and AD-related neurodegeneration. Furthermore, whether the olfactory system has a primary or secondary role in AD is unknown. While the Braak and Del Tredici model suggests a secondary function of the olfactory system changes in preclinical AD, several studies postulate that infections and xenobiotics might drive AD, proposing a primary role.

In the present work, we provide in vivo evidence suggesting that olfactory dysfunction is derived from tau spreading from the brainstem and medial temporal regions and not as an infection gateway from the olfactory system. By combining longitudinal data of odor identification performance, flortaucipir-PET, and diffusion imaging in a well-characterized aging cohort of older individuals –we investigated the pathologic, connectomic, and genetic mechanisms underlying the tau accumulation in the olfactory cortical system. We found a strong relation between olfactory dysfunction and tau accumulation in medial temporal and olfactory regions. Using longitudinal PET data, we found a secondary role of the olfactory system as a receiver from medial

temporal areas. Additionally, we show evidence of several genes spatially impacting AD pathology and olfactory perception.

Across numerous studies, we observed that olfactory dysfunction, measured by UPSIT score, shows reliable associations between cognitive decline and atrophy at the cross-sectional and longitudinal levels. Previous studies on the olfactory system have established a significant loss of olfactory function in normal aging[6], including olfactory abilities like threshold detection, identification, discrimination, and odor memory[11,34]. Smell identification deficits in older adults with healthy cognition at baseline have been shown to predict a 5-year incidence of cognitive impairment[35], and subsequent development of mild cognitive impairment (MCI)[24,36], or progression to dementia[37–39]. Interestingly, the ability of smell deficits to predict cognitive decline has shown to be superior compared to assessments of episodic memory[40]. Structurally, olfactory dysfunction relates to atrophy in olfactory regions[41], entorhinal[23], hippocampus, and amygdala[28,42] and reduced olfactory bulb volume in AD[43,44]; however, it remains unknown

whether the olfactory dysfunction and underlying pathological processes have a primary or secondary role in the etiology of AD. For instance, if the olfactory system deteriorates before the medial temporal structures, it is possible to advocate for its early and critical role in preclinical and clinical AD. We designed the present study to find evidence in this challenging scenario.

While AD is driven by extracellular amyloid-β and intracellular tau accumulation, the latter has shown a more precise association with disease progression and cognitive decline. Neurofibrillary pathology in the entorhinal cortex and CA1/subiculum area of the hippocampus measured postmortem has been associated with odor identification ability ~2.2 years before death[15]. Other early studies in the field found that neurofibrillary tangles are present in the olfactory bulb[45-49] at early Braak stages[22,50] while amyloid appears later with AD manifestation[51]. More recently, lower odor identification ability has been correlated with higher cerebral spinal fluid (CSF) t-tau[52] and p-tau[53] concentrations while not related to amyloid-β biomarkers. Amyloid PET studies found modest or no association with olfactory dysfunction[23,29,54-56]. In this study, we found that a widespread amyloid accumulation pattern was associated with olfactory dysfunction, but this effect disappeared when controlling for tau. On the other hand, the association of olfactory dysfunction with tau accumulation was significant even when controlling for amyloid. Taken together, these findings suggest that odor dysfunction might be a consequence of direct neuronal alterations related to tau pathology.

A recent tau PET study with 26 participants found baseline associations between odor identification changes and tau accumulation in the entorhinal cortex, parahippocampus, fusiform, and inferior temporal gyri[56]. Our study was performed on a larger cohort, finding similar associations at baseline, mainly in the right hemisphere, including the anterior insula, orbitofrontal regions, amygdala, piriform cortex, and anterior olfactory nucleus. While asymmetries in tau pathology accumulation have been repeatedly reported[57,58], it might be driven by the cognitive domain under investigation. This study is based on odor perception, which is more lateralized to the right, while the naming of odor items is usually more related to the left hemisphere[59]. Our brain mapping findings of voxel-level associations between the odor identification task and tau images show a reliable relationship between smell dysfunction and tau accumulation in primary olfactory and medial temporal regions. Moreover, it is worth noting that particular odor items, or linear combinations of them, serve as more specific discriminators associated with early tau stages and tau progression. Odorants such as cherry, leather, and peach have strong weights that can predict tau accumulation in different stages. In contrast, odorants such as motor oil or orange, among others, contribute to the prediction of tau increase in different systems or Braak stages. However, these specific results must be interpreted cautiously because the identification success of individual odor items depends on the odor perceptual features and the response options provided by the UPSIT test. This fact might lead to contradictory results between studies trying to identify specific UPSIT items associated with diseases.

Braak and Del Tredici[20] have suggested that the locus coeruleus is one of the first regions to accumulate tau pathology[21] that later extends toward the trans-entorhinal and entorhinal cortex, hippocampus, and the olfactory bulb[22]. Several studies propose that this spread follows a neuron-to-neuron trans-axonal propagation[20,57,60-62]. Axonal integrity from locus coeruleus to trans-entorhinal is significantly decreased in AD[63] and increased with CSF tau levels[64]. In the present study, we combined tau imaging with tract integrity analysis to study the pathology progression in the olfactory system, including projections of the medial temporal lobe and brain stem, in order to cover a complete scenario of potential spreading pathways of tau. Using a graph theory approach based on longitudinal PET data[31], we found that medial temporal lobe structures, such as the amygdala and the entorhinal cortex, are central in disseminating tau toward

olfactory-related regions – particularly the piriform cortex. Moreover, our findings show that projections from the LC entering the raphe nucleus and reaching trans-entorhinal regions are also associated with olfactory dysfunction and may be related to the mechanism underlying tau spreading from the brainstem toward the medial temporal lobe. Together, these connectivity and tau patterns suggest that the olfactory system accumulates pathology as receiver regions from the medial temporal lobe and/or the brainstem, rather than acting as a primary source of tau spreading.

The elucidation of the primary or secondary role of the olfactory system changes in preclinical and clinical AD is essential. In the past, it has been postulated that transnasal infectious or environmental factors might have causal implications for AD pathology[65-67]. For instance, some evidence suggests that nasal epithelial infections could activate the brain's immune system, leading to neuroinflammation and the production of amyloid[68], which is thought to be a natural antimicrobial protection in the brain[69]. Our results showed that tau is critical in driving the association with olfactory deficits. Moreover, we found widespread amyloid (not specific to olfactory regions) also associated with olfactory scores and might be synergistic with tau accumulation. In addition to previous studies describing a relationship between tau and olfactory identification only in amyloid-positive cases, these results might partially support this idea. It can be speculated that reactive amyloid accumulation due to xenobiotics or respiratory infections could facilitate the appearance and spreading of tau in the olfactory system, accelerating neurodegeneration. However, in our study, the association between amyloid and olfactory scores did not survive when controlling for tau, leading to the interpretation that tau must be a fundamental independent factor of olfactory dysfunction in aging. Thus, our findings do not support the hypothesis that tau spreads from olfactory regions to other brain regions, as it would be expected if the olfactory organs were the infection gateways and first in the tau spreading chain of events. Instead, it supports that tau accumulation is an intrinsic factor of neurodegeneration in the limbic-to-olfactory pathway.

Neuroimaging-genetic studies of in-vivo PET imaging and postmortem brain transcriptome atlases[70] have enhanced our understanding of biological factors underlying brain phenotypes of complex neurodegenerative diseases such as AD[31,71]. In this study, we investigated the spatial intersection of tau spreading patterns and expression levels of genes related to odor perception to investigate why the olfactory system seems to have an intrinsic vulnerability to developing AD pathology. Genetic variations in olfactory receptors have been associated with many other body functions[72] and diseases, including earlier onset of AD[73]. Single-cell sequencing in the olfactory mucosa also showed different regulated genes in AD patients[74]. These findings suggest a link between tau co-expression and odor perception genes might exist. Evidence from the present study supports that view. We found that there are spatial brain distributions of olfactory perception genes similar to tau accumulation maps in the olfactory system. This study shows several genetic variants impacting olfactory genes could be associated with tau accumulation, amyloid, Alzheimer's disease, aging, cognition, and brain morphometry. Associations between olfactory genes and tau accumulation are more abundant and robust than those with amyloid accumulation. Interestingly, the biggest olfactory co-expression cluster showed higher expression in the right hemisphere. Moreover, we described that one gene stands out, namely, *TOMM40*, that was associated with all the domains. In particular, the rs2075650 (*TOMM40* intron variant in chromosome 19 and position 44892362) has been associated with smell perception[75], tau and amyloid accumulation, Alzheimer's disease, aging, cognitive decline, and brain atrophy.

Limitations of this study should be noted. First, although we use strict masking of the data, it is essential to consider that Flortaucipir (FTP)-PET data have off-target binding[76], particularly in choroid plexus

and neuromelanin-rich areas. Second, our investigation of spatial cortical integration is based on a neuroimaging-genetics approach that uses transcriptomic atlas data from the AHBA generated with six postmortem brains. We did not include individual genotyping or brain transcriptomic of the HABS participants. Therefore, only group-level and not individual-level interpretations can be drawn from our neuroimaging-genetic study. The results must be considered preliminary and should be replicated with a serial assessment of both olfaction and tau PET imaging at multiple time-points. Lastly, our sample is constrained by healthy aging individuals with UPSIT baseline measures without information about past viral nasal infections.

In summary, our study offers several observations about the relationship between odor discrimination changes, neurodegeneration, connectomic patterns, and tau spreading pathways in the olfactory system of the human brain during aging. The study's findings suggest that tau aggregates in the olfactory system, predominantly due to earlier tau accumulation in the medial temporal regions. This results in olfactory dysfunction years before any appreciable cognitive symptoms. The spatial brain expression of odor perception genes and its co-expression with tau-related genes advocates a common background that deserves further exploration to elucidate potential mechanisms of neuronal vulnerability in olfactory and limbic-related neurons.

## Methods

### Participants
We included 418 clinically normal participants from the Harvard Aging Brain Study (HABS; a longitudinal study on aging and AD)[30] with smell identification measures (72 ± 8.34 years; 251 females/167 males; see Supplementary Table 1 for detailed demographics). One hundred fifty-five of these participants also had a flortaucipir PET tau acquisition within six months of the smell test (71.05 ± 10.02 years; 85 females/70 males; see Supplementary Table 2 for detailed demographics), out of which 89 have a second flortaucipir PET tau acquisition ~2.5 years after the baseline (65.88 ± 8.43 years at baseline; 2.38 ± 0.45 years between 2 scans; 50 females/39 males; see Supplementary Table 3 for detailed demographics). Additionally, we studied 82 individuals of this sample with available diffusion tensor imaging (73.84 ± 8.30 years; 48 females/ 34 males; see Supplementary Table 4 for detailed demographics). Inclusion criteria in HABS included participants having a score of 0 on the Clinical Dementia Rating scale, an MMSE score ≥ 25, and performing within education-adjusted norms on the logical memory delayed recall test (> 10 for ≥ 16 years of education, >6 for 8–15 years of education and > 4 for <8 years of education). Participants with clinical depression (Geriatric Depression Scale above 11/30) or other psychiatric illnesses, history of alcoholism, drug abuse, head trauma, or a family history of autosomal dominant AD dementia were excluded from the study. Data was acquired before the SaRS-CoV-2 outbreak, so COVID-19 was not a potential confounder of odor loss. The study complied with all ethical regulations and was approved by the MGB/ Partners Human Research Committee at Massachusetts General Hospital (IRB Protocol #: 2010P000297). All participants provided written informed consent following the MGB/Partners Human Research Committee regulations and received monetary compensation after each visit.

### Olfactory testing
We used the University of Pennsylvania Smell Identification Test (UPSIT)[34] scores to quantify odor identification performances at baseline and follow-up visits. This test is a non-invasive approach that contains 40 microencapsulated common odor items to scratch and sniff. The identification of each of these odorants is performed by choosing from 4 alternative choices. The composite score ranges from 0 to 40 depending on the correct identification of odor items.

### Cognitive assessment
The cognitive performance of the participants was measured with the Preclinical Alzheimer Cognitive Composite (PACC)[77], a multi-cognition domain composite developed to improve the detection of preclinical AD and weighted towards memory performance. This composite is comprised of the delayed recall score of the logical memory subtest (Weschler Memory Scale), the total score of the Mini-Mental State Examination (MMSE), the digit symbol substitution score of the Wechsler Adult Intelligence Scale Revised (WAIS-R), and the free and total scores of the free and cued selective reminding test (FCSRT). The composite averages the z-transformed values of each variable based on HABS baseline mean and standard deviation data.

### Structural Magnetic Resonance Imaging
A Siemens 3 Tesla Tim Trio with 12-channel phased-array head coil was used to acquire structural images. The protocol included a T1-weighted magnetization prepared rapid gradient-echo (MPRAGE) scan with the following parameters: repetition time (TR) = 6400 ms, echo time (TE) = 2.8 ms, flip angle = 8°, inversion time (TI) = 900 ms and a voxel size of 1.0 × 1.0 × 1.2 mm. Additionally, a single-shot spin echoplanar sequence was used to acquire diffusion-weighted images with the following parameters: repetition time (TR) = 8,040 ms, echo time (TE) = 84 ms, flip angle = 90°, 2 mm isotropic voxel size, 30 isotropically distributed gradients with a b-value of 700 s/mm2 and five non-diffusion weighted images ($b = 0$ s/mm2)). MRI data were preprocessed using FMRIB Software Library v5.0.7 (FSL). To study volumetric differences in the brain, the anatomical T1 preprocessing pipeline included: reorientation to right-posterior-inferior (RPI); alignment to anterior and posterior commissures; skull stripping; and gray matter (GM) segmentation. An optimized voxel-based morphometry was used to assess grey matter volume changes[78]. The partial GM volume estimations were transformed to 2 mm MNI 152 standard space using non-linear registration. The resulting images were averaged and flipped along the x-axis to create a left–right symmetric, study-specific gray matter template. Then, all native gray matter images were non-linearly registered to this study-specific template and modulated to correct for local expansion (or contraction) due to the non-linear component of the spatial transformation. Finally, we smoothed the modulated gray matter images with an isotropic Gaussian kernel with a sigma of 3 mm (7 mm full width at half maximum).

Diffusion imaging preprocessing included: eddy current correction; gradient vector rotation to compensate for head motion; local fitting of the diffusion tensor at each voxel; and computation of Fractional Anisotropy (FA) and Mean Diffusivity (MD) maps. The transformation between diffusion and T1-weighted images was computed and concatenated with the computed transformation between T1 and MNI standard space T1. To model crossing fibers, we used the FSL BEDPOSTX tool with default parameters, and subsequently, the PROBTRACKX2 tool taking 1000 samples from the range of possible principal diffusion directions within each voxel to compute the path of fibers connecting the locus coeruleus with medial temporal and primary olfactory regions: anterior olfactory nucleus (AON), olfactory tubercle (TUR), piriform cortex (Pir), entorhinal, amygdala and parahippocampal cortex. We used the AON, TUR, and Pir regions from Zhoe et al[79]. and locus coeruleus from Eckert Lab[80]. We also employed the Freesurfer segmentation and Desikan-Killiany parcellation schemes for the entorhinal, amygdala, and parahippocampal regions. Additionally, the probabilistic olfactory tract from Echevarria-Cooper[81] et al. was used to study microstructural properties in the olfactory tract.

### Flortaucipir (FTP)-PET
Fluorine 18-FTP PET images were acquired in a Siemens/CTI (Knoxville, TN) ECAT HR+ scanner with the following parameters: 10 mCi of 18F-T807, 3D-mode static protocol of 20-min acquisition from

80–100 min; 63 image planes, 15.2-cm axial field of view, 5.6-mm transaxial resolution and 2.4-mm slice interval; 4 × 300-s frames at 80 min post-injection followed by a 6-min transmission[82]. The resulting PET-acquisition was reconstructed, attenuation corrected, and quality checked for head motion. We rigidly co-registered PET images using subjects' MPRAGE and spatially normalized them into MNI standard space using SPM12 (Welcome Department of Cognitive Neurology, Function Imaging Laboratory, London). We processed T1-weighted images in FreeSurfer (FS) version 6 and used the three-compartment model (Muller-Gartner MG Method) for partial volume correction. Last, we computed the standardized uptake value ratio (SUVr) using FS's cerebellar gray as the intensity reference region.

## PiB-positron emission tomography

Carbon 11–PiB was acquired on a Siemens ECAT EXACT HR+ system following a transmission scan. 10–15 mCi 11 C-PiB was injected intravenously as a bolus and followed immediately by a 60-min dynamic PET scan in 3D mode (63 image planes, 15.2-cm axial field of view, 5.6-mm transaxial resolution and 2.4-mm slice interval; 69 frames: 12 × 15 s, 57 × 60 s). PiB retention intensity was expressed as the distribution volume ratio (DVR) using Logan's graphical method and the cerebellar gray region as reference. Corrections for normalization, dead time, random coincidences, scattered radiation, and attenuation were performed, and each frame was evaluated to verify adequate count statistics and the absence of head motion. Subjects MPRAGE was used to spatially normalize PET data into MNI standard space using SPM12, and the three-compartment model (Muller-Gartner MG Method) from FreeSurfer was used for partial volume correction.

## Characterization of Odor Identification in Aging

We used the UPSIT assessments of a cross-sectional sample of 418 cognitively normal individuals from HABS to evaluate olfactory deficits. Robust regression was used to model odor identification changes in healthy aging as a quadratic function using Random Sample Consensus (RANSAC) controlling for outlier participants (Fig. 1a). Five thousand iterations with 20 random participants per iteration were used to obtain the model with the best fit (major number of inliers; 6 deviation points in the smell identification test of the predicted UPSIT value was used to differentiate inlier and outlier participants). We also computed the percentage of subjects with correct odor identifications to measure the ability to discriminate each of the 40 odorants composing the UPSIT test (Fig. 1b).

## Tau and Neurodegeneration Associations with Olfactory Dysfunction

We used the tau-PET and structural T1 images of 155 individuals within six months of the olfactory measurement. For each of the 40 different odorants, we performed a voxel-wise regression analysis comparing tau accumulation between the group of subjects that could identify the odor with those who couldn't – controlling for age, sex, and smoking history ($p$-value < 0.05; Monte-Carlo simulation to correct for multiple comparisons; Fig. 1c). Five thousand iterations were used to generate 40 random maps in each iteration with similar smoothness as the original data residuals. These random maps were used to quantify the likelihood of a voxel being significant for x number of tests with a p-value < 0.05 by chance. This approach was repeated with voxel-based morphometry data to study whether olfactory tests could be associated with atrophy in the brain (Fig. 1d).

## Olfactory Measures and Cognitive Prognosis

We used a general linear model – controlling for age, sex, years of education, and smoking history – to study the association of odor identification measures (UPSIT) with cognitive scores (PACC) and with tau images both at baseline and ~2.5 years after the olfactory testing ($N = 89$; Fig. 1e). For the longitudinal analysis, the general linear model

additionally controlled for the time difference between the two PET scans. All findings were corrected for multiple comparisons using Monte Carlo simulation (clusterwise correction) with 10,000 iterations to estimate the probability of false positive clusters with a p-value of <0.05. The analysis was repeated, adjusting for voxel-level volume and amyloid accumulation.

## Olfactory biomarkers maximizing prediction of protein accumulation

We performed a graph theory approach to evaluate the best olfactory biomarkers – the best linear combination of individual odorant identification measures (see Fig. 3). For each of the 40 odorants, we compute a voxel-wise association analysis comparing tau accumulation between the subjects that could identify the odor with those who could not – controlling for age, sex, and smoking history ($N = 155$). We repeated the same association analysis with amyloid accumulation. Then, we generated a network of 40 odorants x (tau voxels + amyloid voxels) with the Z-statistic of the association between each pair. Next, principal component analysis was used to reduce all the information in the matrix in components – linear combinations of odorants that maximize the association between olfactory dysfunction and voxel-level protein accumulation in the whole brain. We used the weights given to each odor item to compute an olfactory dysfunction score for each individual. We calculated a voxel-wise regression of the obtained olfactory dysfunction scores with voxel-level tau and amyloid accumulation, controlling for age, sex, and smoking history (see Fig. 3a). Findings were corrected for multiple comparisons using Monte Carlo simulation (clusterwise correction) with 10,000 iterations to estimate the probability of false positive clusters with a p-value of <0.05. The same approach was repeated to find odor identification biomarkers that predict tau increase (difference between longitudinal and baseline tau); in this case, using a 40 odorants x tau difference voxels matrix ($N = 89$).

## Microstructural properties on olfactory pathways

We used diffusion tensor imaging to investigate the white matter microstructural integrity of the olfactory tract and the tract connecting locus coeruleus with the medial temporal lobe and olfactory system and their relationship with the UPSIT performance. First, a skeletonization algorithm was used to shrink both tracts and divide them into 100 contiguous same-size segments. All the voxels belonging to each tract were assigned to the closest segment. For each participant, the average Mean Diffusivity (MD) of the voxels belonging to each segment was computed as a proxy of microstructural damage. Then, we used a general linear model to compute the association between the MD in each tract segment and the UPSIT test, controlling for age, sex, and smoking history ($N = 82$; Fig. 4a). We projected these results as line plots where the significance of UPSIT associations (y-axis) and distance from the source, as well as correspondence with cortical regions (x-axis), can be straightforwardly visualized (Fig. 4a). Additionally, the MD differences in the whole cortex were evaluated and corrected for multiple comparisons using Monte Carlo simulation (cluster-wise correction) with 10,000 iterations to estimate the probability of false positive clusters with a p-value of <0.05.

## Directional graph theory regression of tau accumulation

In contrast to conventional analysis approaches in PET imaging examining regional binding differences, we used a graph theory approach to identify the directionality of tau spreading using longitudinal tau connectivity[31]. Tau connectivity measures how the increase of tau in a region is associated with an increase in another region[83]. Further extending this approach to longitudinal data, we can study how tau at baseline is associated with longitudinal tau increase in any other brain voxel. To minimize cross-sectional associations and look to associations with the longitudinal increase in tau, we controlled the

longitudinal connectivity by the tau accumulation at the baseline of the evaluated longitudinal region. Compared with previous approaches, this correction assures the connectivity reflects longitudinal spreading and not tau accumulation covariance. First, we computed the average tau intensity for each region of interest and time point at the individual level. Then, to construct bipartite graphs of tau data and assess spreading directionalities, we used the piriform cortex, olfactory tubercle, anterior olfactory nucleus, dorsal raphe nucleus (DRN), entorhinal, amygdala, and parahippocampus regions as nodes. For each pair of $i, j$ regions, the correlation of participants' baseline tau at region $i$ with longitudinal tau at region $j$ was computed controlling for baseline tau at region $j$ (Fig. 4b). Thus, we obtained an asymmetric connectivity matrix ($p$-value < 0.05 level). The mean diffusivity data of DRN was also included in the analysis.

### Integration of diffusion pathways and Tau spreading patterns

We used the skeleton identification step of the PC-algorithm[84], an independence-based causal discovery algorithm, to identify the main backbone of tau progression in the olfactory system. While the bipartite graph approach displays indirect connections due to a common neighbor between two nodes, this algorithm uses conditional independence testing to obtain a network without indirect connections. The algorithm starts with a complete undirected graph between all regions of interest and removes all the edges $X - Y$ where $X$ and $Y$ are independent given a set of variables $Z$. Initially, $Z$ is an empty condition set that will increase the number of variables. Tau accumulation at baseline and MD in dorsal raphe nucleus was used to derive the backbone of tau propagation (Fig. 4c).

### Gene co-expression analyses and associated neurodegenerative traits

We used the Allen Human Brain Atlas (AHBA) to identify the group of genes related to sensory perception of smell with a spatial brain expression resembling tau propagation in the brain. AHBA is a publicly available brain transcriptome database, providing genetic expression of 20,737 protein-coding genes extracted from 58,692 measurements in 3702 brain samples obtained from 6 individuals[70]. We employed a projection transformation of the samples to the 68 cortical regions covering the whole cortex of the Desikan-Killiany atlas[85]. After mapping each sample to a cortical region of the Desikan-Killiany atlas, we computed the average genetic expression across all samples mapped into a specific region for each donor. Finally, we calculated the median values between the gene expression of the six donors, and then each gene map was z-scored[86]. Three hundred ninety genes annotated as sensory perception of smell between gene ontology (GO:0007608) and GWAS catalog[87] were used to compute a gene-by-gene co-expression matrix. We used Pearson correlations to assess the co-expression of pairs of genes based on the brain spatial distribution of its expression (Fig. 5). A Group of co-expressed genes were identified using agglomerative hierarchical clustering, and the Silhouette method was used to evaluate the optimal number of clusters. GeneCards Version 5.11[88] was also used to obtain traits associated with each gene and reported in GWAS Catalog, a quality-controlled and manually curated catalog including GWAS studies assaying at least 100,000 variants and with $p$-values < $1.0 \times 10^{-5}$. Reported associations by GeneCards included the source SNP and its mapped gene exons and GeneHancer[89] regulatory elements. Only traits associated with tau, amyloid, Alzheimer's Disease, aging, cognitive decline, and brain morphometry were selected for further analysis. For each group of olfactory genes, a bipartite graph was created between the genes and traits of interest. The weight of the connection was defined with the best p-value obtained from GWAS -the best p-value of the association of a trait with an SNP associated with a particular gene-. For a more compact representation, the traits were grouped into six domains: tau, amyloid, Alzheimer's disease, aging, cognition, and

brain. Once again, the best score of the graph was used to obtain the reduced network.

### Reporting summary

Further information on research design is available in the Nature Portfolio Reporting Summary linked to this article.

## Data availability

All neuroimaging and clinical data supporting this study's findings are available from https://habs.mgh.harvard.edu/researchers/request-data/. HABS data curation is overseen by Aaron P. Schultz (aschultz@nmr.mgh.harvard.edu) at the Martinos Center for Biomedical Imaging, Massachusetts General Hospital, Harvard Medical School, Boston, MA. The Allen Human Brain Atlas (AHBA) transcriptomic data is available at https://human.brain-map.org. Source data used in the figures are provided as a Source Data File. Source data are provided with this paper.

## Code availability

All codes related to imaging analysis are available for the research community from the corresponding author (J.S.) upon request for the purpose of scientific investigation, teaching, or the planning of clinical research studies.

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

## Acknowledgements
We thank the investigators and staff of the Harvard Aging Brain Study, Massachusetts Alzheimer's Disease Research Center, the individual research participants, and their families and caregivers. We also thank the Gordon Center for Medical Imaging, the PET Core of the MGH, the Harvard Center for Brain Science Neuroimaging Core, and the Athinoula A. Martinos Center for biomedical imaging support. This research was partly made possible by the computational hardware generously provided by the Massachusetts Life Sciences Center (https://www.masslifesciences.com/). This research was supported by grants from the National Institutes of Health (NIH) (R01AG061811 to J.S.; R01AG061445 to J.S.; R01AG061083 to J.S. and P.V.; R01HL137230 and P41-EB022544 to G.E.F.; R01-AG027435-S1 to R.A.S.; P50-AG00513421 and R01AG046396 to R.A.S.; P01-AG036694 to R.A.S.; R01AG062559, R01AG068062, R21AG074220, R01AG082006 to H.I.L.J.; U01DC019579 to M.W.A), The Massachusetts General Hospital Executive Committee On Research (ECOR) Interim Support Funding (ISF) for I.D., the Alzheimer's Association (AARF-23-1145358 to I.D and J.S) and the Swedish Research Council (2020-00266 to J.O.). M.W.A. is a co-founder and owns shares in Aromha, Inc. The other authors declared no potential conflicts of interest concerning this article's research, authorship and/or publication.

## Author contributions
I.D. and L.O.T. contributed to the study design, methods development, imaging data and statistical analyses, data interpretation, and writing of the manuscript. T.S.C.N. contributed to data analysis, interpretation, and manuscript preparation. M.W.A. and G.M. participated in the data collection, data analysis, and interpretation of the data. W.O., C.K., E.B., V.M., J.O., P.V., G.E.F., R.A.S., K.J., H.I.J., J.S. conceptualized this study, participated in study design, methods development, statistical analyses, interpretation of the data, preparation of the manuscript, and had the general supervision of the study.

## Competing interests
M.W.A. is a co-founder and owns shares in Aromha, Inc. The other authors declared no potential conflicts of interest with respect to the research, authorship and/or publication of this article.
