## [Peer Review File · Nature Communications]

Tau propagation in the brain olfactory circuits is associated with smell perception changes in agingReviewer #1 (Remarks to the Author):

This is a clinical research study on olfactory identification deficit and brain changes using the Harvard Aging Brain Study.

While the data is interesting, it is mostly confirmatory. The association of OID with cognition, structural MRI changes in the olfactory pathway and connectivity have all been published.

As this is an association study, it does not prove etiology, and does not address infectious etiology, that is an overstatement.

Of note, the relevant question for OID is whether it is useful as a non-invasive biomarker differentiating normal aging from preclinical AD; is the mechanism of OID different in aging versus neurodegeneration.

The authors find correlation between tau pathology localization and OID, which may reflect the subthreshold, clinically not detected, neurodegeneration. This is likely, as there is a relationship with the cognitive measure. These data indicate that the earliest tau neurodegeneration associated OID originates from the entorhinal cortex and mesial temporal structures, which has been known. The question is where the aging associated OID localizes and are there specific odors that can distinguish between the normal aging versus neurodegenerative process.

Fig 3 suggests stage independent and stage dependent smell loss, which is confirmation of previous studies. The odors depicted in year 2.5 cov baseline are mostly irritants. It has been shown that these irritant odors are not pure olfactory chemosensation, and tissue mechanoreceptors could facilitate perception of these smells after olfactory pathway impairment.

Fig 5 is data mining of target genes that have been associated with the olfactory system. TOMM40, the gene in linkage disequilibrium with APOE4, emerged as being associated with tau neurodegeneration. APOE4 is the undisputed genetic association for AD pathology and the models for this study have not incorporated APOE genotype. This raises the question whether the OID and tau association is driven by the latent association with APOE4. All statistical models should be rerun with APOE4 genotype in the model.

Reviewer #2 (Remarks to the Author):

This interesting and ambitious study sought to examine, in a population of older persons, associations between differences in olfactory function, as measured by a well-validated smell test, and patterns of brain neurodegeneration and connections among brain structures, as well as the spreading of tau within selected olfaction-related brain regions. The authors suggest that tau accumulation within the medial temporal region is associated with early stages of Alzheimer's disease, resulting in smell loss observed at that time. Co-expression of odor- and tau-related genes is noted. The general conclusion is that tau emerges from the limbic system and only secondarily involves olfactory structures that would be implicated if an exogenous agent entered the olfactory pathways from the nose. The work is largely novel and appears to be well performed.

Comments:

Abstract: The abstract could benefit with more detail – numbers of subjects, age range, procedures, etc. Presently it does not provide even minimal description of what was actually done.

Introduction, p. 4, lines 51-7: While the olfactory system is unique in terms of largely bypassing the thalamus before reaching cortical regions, the other major sensory systems also involve the entorhinal cortex and the locus coeruleus. Hence olfaction is not unique in this regard. This section should be reworded accordingly.

p. 4, line 19. Perhaps want to also consider other xenobiotics and reword something like, "...a link with viral and other xenobiotic impacts on the olfactory pathways." I am surprised that none of the extensive work of Calderón-Garcidueñas et al. is considered or even mentioned in the introduction or discussion (e.g Calderón-Garcidueñas et al., Urban air pollution: Influences on olfactory function and pathology in exposed children and young adults. *Experimental and Toxicological Pathology*, 62:91-102, 2010). See also O'Piela et al., Particulate matter and Alzheimer's disease: an intimate connection. *Trends in Molecular Medicine* 2022 Sep;28(9):770-780. Wouldn't these types of

studies add potential fuel to the concept of entrance of agents into the olfactory system relative to AD? How do Calderón-Garcidueñas et al.'s results fit with the data and the general theory subscribed to in the present study? Does tau have to be the final arbiter?

Results, p. 7, lines 15+; p. 9, lines 8-15. Attempts to identify specific UPSIT items involved in a number of neurodegenerative diseases have been contradictory and have shed little light on the association between smell and such diseases. It is not clear that seeking to identify specific UPSIT items relative to tau or Braak stages has much scientific value. Indeed, earlier studies have lacked uniformity of findings across UPSIT items (e.g., for Parkinson's disease). This reflects, in part, the fact that the different UPSIT stimuli are not iso-intensive and many are individually made up of numerous chemical compounds. Hence, names like "rose", "dill pickle" etc. would not appear to have much operational scientific value. It appears that the UPSIT stimuli that were most correctly identified in this study were the more intense odors. This problem is complicated by a number of other factors. For example, receptor types of unknown ratios are present in the epithelium, some of which are responsive to more than one defined chemical, i.e., are so-called generalists. Cells expressing the same receptor type may have different tuning curves, and the dynamic ranges of various chemicals vary. Hence, my suggestion would be to either address these issues in the discussion or omit these elements from the paper.

Results, p. 8, lines 12+. Why was smell only performed at baseline? While it is interesting to determine whether the baseline UPSIT score could predict tau progression over time, it would seem similarly interesting to determine whether the smell test score changed in the same manner as the change in tau. Perhaps this could be addressed in the discussion section.

Results, p. 9-11. It appears that a number of assumptions have to be made in the various analyses in order to come to the conclusions. Mentioning these assumptions may be helpful to readers who lack detailed knowledge of these types of analyses. For example, can one truly infer that by measuring the amount of tau at two points in time in different brain regions that tau is spreading between them? According to the title, Tau is propagating through olfactory brain circuits. Are there alternative explanations, such as development of differential sensitivity of brain regions to tau formation over time?

It would be helpful to have a paragraph at the end of the discussion outlining the strengths and limitations of various elements of the study. It is my understanding, for example, that there are limitations of using gene expression data in attempts to identify functional associations and perhaps this can be articulated in this paragraph.

Reviewer #3 (Remarks to the Author):

Olfactory deficits are known to be related to tau pathology in the brain, particularly in Alzheimer's disease and Lewy body disease. The authors examine associations between individual items on the UPSIT and Flortaucipir tau PET imaging. While this is an area definitely worth investigating, the results do not answer the main questions in a clear and logical manner.

1. There were 418 cognitively normal individuals studied, 155 of them got tau imaging at baseline and 89 at follow-up an average of 2.5 years later. There is no clear explanation of which of these samples were examined in which analysis in the text nor in the figure legends. Baseline cross-sectional analyses are not distinguished from longitudinal analyses in drawing the conclusions that are made. This is important because these were cognitively normal individuals and not much change in olfaction or tau imaging would be expected over 2.5 years.
2. The claim that olfaction is related to tau PET and not amyloid PET indices can be ascertained by a simple analysis involving these three measures, but this is not directly described.
3. Flortaucipir shows off-target binding with PET. There is no discussion of this issue or the limitations of the study, which include only 21% of the original sample getting follow-up tau PET scans. There is no description of how the missing data were handled.

4. The UPSIT total score is the conventional measure to analyze for this test. The correlation between the total UPSIT score and the tau binding (total and ROIs) is not presented, suggesting that it was not robust enough to describe. Instead, the correlations for each of 40 UPSIT odorants with tau imaging parameters is the focus of the paper.
5. Did baseline total UPSIT correlate with baseline tau imaging parameters (total binding, ROI binding) and did it predict change in tau imaging parameters? Did the change in total UPSIT total score track directly with the change in tau measures over the 2.5 years of follow-up? These results are essential (but missing) to address the questions posed by the authors. The associations described with sophisticated image analysis are difficult to interpret in the absence of a description of the correlation coefficients among the main measures of interest.
6. Which covariates were used? Do the findings remain after covarying for age?
7. A quadratic rather than a linear function was used. No explanation is provided for this choice.
8. A novel graph theory approach was used. There is no description of what this involves and whether any validation has been published.
9. Overall, the analyses are based on a large number of assumptions without the presentation of basic statistics about the associations between the variables of interest and how they change over time.
10. Genes derived from a dataset in the literature from 6 individuals are analyzed. These analyses do not seem to be related to the main point of the paper regarding the associations between olfaction and tau imaging.

Reviewer #1

This is a clinical research study on olfactory identification deficit and brain changes using the Harvard Aging Brain Study.

We thank the reviewer for taking the time to thoroughly review the manuscript and for the helpful suggestions to improve it. We believe these edits have substantially improved the quality of the manuscript.

While the data is interesting, it is mostly confirmatory. The association of OID with cognition, structural MRI changes in the olfactory pathway and connectivity have all been published.

We thank the reviewer for raising this point regarding the confirmatory nature of these findings integrating OID, cognition, and structural MRI. It is worth noting that the associations of OID with cognition and MRI only served us as the starting point for this project. We included Figure 1 (describing the association between cognition and structural MRI) to illustrate the characteristics of our sample.

Our investigations on tau PET and neuroimaging-genetics integration in the olfactory pathway are novel and unprecedented. These sections of the manuscript are the core of highly innovative findings. The cross-sectional and longitudinal tau imaging results provide new mechanistic insights about how tau pathology spreads to olfactory regions from medial temporal areas, as well indicate the significant role played by amyloid in accelerating that process (see revised Figure 3 and Supplementary Figure 2). Furthermore, we report: 1) new OID-related biomarkers that maximize the prediction of cross-sectional and longitudinal tau and amyloid accumulation; and 2) candidate genes associated with the impact of the olfactory pathway in aging, opening new opportunities to identify the vulnerability of specific brain systems to develop AD-related pathology, both of which are novel results in the field.

To illustrate how innovative this work is, from an independent source, we would like to note that it has been recently presented and recognized as one of the plenary sessions at the HAI conference (Miami, January 2023) -the most important international venue for molecular imaging in aging and Alzheimer's disease research- (please see the 2023 HAI program: <https://hai.worldeventsforum.com/program/>; "Tau propagation in the brain olfactory circuits" presented by Dr. Ibai Diez).

As this is an association study, it does not prove etiology and does not address infectious etiology, that is an overstatement.

We agree with the reviewer that these analyses do not use a direct causal or experimental intervention to investigate the relationship between tau spreading and previous infections or xenobiotics exposure. However, causal connectomic algorithms are able to establish temporal relationships between nodes[1,2]. These structure learning algorithms in probabilistic graphical models can estimate causal relationships from longitudinal data. In our case, we used conditional independence testing to study causal relationships in the spreading of tau. Our findings showed a robust spatiotemporal progression from medial temporal to olfactory regions and not the contrary direction (olfactory-to-medial-temporal). Given that it is plausible to speculate that external agents would build up tau in the olfactory-to-medial-temporal pathway direction, these results support that agents that may use the olfactory entrance gateway are less probable than

internal brain factors. Following the reviewer's comment, we have modified the main text to clarify this point accordantly.

[1] Sepulcre, J. et al. Neurogenetic contributions to amyloid beta and tau spreading in the human cortex. *Nat Med* 24, 1910–1918 (2018).

[2] Krance, S. et al. Reciprocal Predictive Relationships between Amyloid and Tau Biomarkers in Alzheimer's Disease Progression: An Empirical Model. *J Neurosci* 39, 7428-7437 (2019).

Of note, the relevant question for OID is whether it is useful as a non-invasive biomarker differentiating normal aging from preclinical AD; is the mechanism of OID different in aging versus neurodegeneration. The authors find correlation between tau pathology localization and OID, which may reflect the subthreshold, clinically not detected, neurodegeneration. This is likely, as there is a relationship with the cognitive measure. These data indicate that the earliest tau neurodegeneration associated OID originates from the entorhinal cortex and mesial temporal structures, which has been known. The question is where the aging associated OID localizes and are there specific odors that can distinguish between the normal aging versus neurodegenerative process.

We thank the reviewer for raising this comment and agree that the role of OID as a potential biomarker for differentiating normal aging from preclinical AD is an important question. Following the reviewer's suggestion, we use tau and amyloid burden to differentiate preclinical AD vs. normal aging and perform a new set of analyses to detect the best biomarkers - the best linear combination of UPSIT odors - that maximize the explained variance of voxel-level tau and amyloid accumulation. First, we computed the association of every voxel tau with olfaction and every voxel amyloid with olfaction adjusting for age, sex, and smoking history (N=155; Figure 3). We applied principal component analysis to reduce the amount of information and study the component maximizing the explained variance between the association of olfactory deficits with voxel-level tau and amyloid accumulation. Using this strategy, we obtained an OID biomarker associated with tau accumulation in medial temporal and olfactory regions and widespread amyloid accumulation (Supplementary Figure 2). Widespread amyloid in these regions has been associated with preclinical AD. We additionally search for other OID biomarkers – different linear combinations of odorants – that could predict the increased tau accumulation in different brain systems or Braak stages. Figure 3c shows the cortical projections of where the 3 longitudinal biomarkers could predict tau increase, and Figure 3a shows the linear combination of odors leading to these biomarkers. In light of these new results, our sample shows that 1) tau associations with olfactory systems are more specific than amyloid accumulations (please see also Figure 3 regarding the independent and dominant effect of tau), and 2) it represents an aging cohort in which is difficult to separate the “normal aging” and “preclinical AD” component due to the presence of amyloid deposits in these individuals.

Supplementary Figure 2

Regions of increased tau and amyloid accumulation explaining most of the variance associated with identification of 40 individual odorants

Supplementary Fig. 2 | Voxel-level association between tau and amyloid PET with a linear combination of individual UPSIT items that maximized voxel-level tau and amyloid prediction.

Fig3 suggests stage independent and stage dependent smell loss, which is confirmation of previous studies. The odors depicted in year 2.5 cov baseline are mostly irritants. It has been shown that these irritant odors are not pure olfactory chemosensation, and tissue mechanoreceptors could facilitate perception of these smells after olfactory pathway impairment.

Thanks to the reviewer's comment, we have now investigated the combinatory/grouping effects in more detail in the revised manuscript. Moreover, we would like to note that another reviewer also pointed out that different UPSIT stimuli are not isointense, and many are individually made up of numerous chemical compounds. Other factors might intervene in the single-odor perception task in research studies as well: (i) receptor types of unknown ratios present in the epithelium, some of which are responsive to more than one defined chemical, i.e., are so-called generalists; (ii) cells expressing the same receptor type may have different tuning curves, and the dynamic ranges of various chemicals vary; etc. These points motivated us to move Figure 3 to supplementary material (Supplementary Figure 4) and create a new Figure 3 looking at linear combinations of odorants that maximize voxel-level tau and amyloid increase prediction. We examined how different linear combinations of scents could predict tau increase (difference from longitudinal and baseline PET) in different brain regions (Figure 3c). Compared to the previous approach, we can observe the contributions of all odors to the cross-sectional and longitudinal spreading of tau. We added Supplementary Figure 3, showing the Jaccard similarity index between scents (N=418) to fully describe these results.

Fig. 3 | Olfactory biomarkers for tau accumulation and longitudinal increase. (A) Linear combination of 40 odorants (olfactory biomarkers) maximizing explained variance of cross-sectional voxel-wise amyloid and tau accumulation prediction (Cross1; N=155). In different brain circuits, three other linear combinations predict longitudinal tau increase in 2.5 years (Long1-3; N=89). (B) Z-stat of the association of Cross1 olfactory biomarker and tau accumulation in the brain. Results survived when adjusting for amyloid. See supplementary figure 2 for association with amyloid. (C) Three components - different odorant linear combinations – predict longitudinal tau increase in the regions projected on the surface. Only areas surviving multiple comparisons when computing the association between tau accumulation and olfactory biomarkers are displayed. All the previous associations were adjusted for age, sex, and smoking history.

Supplementary Fig 3

Supplementary Fig. 3 | Similarity (Jaccard index) between the UPSIT items identification ability in N=418.

Figure 5 is data mining of target genes that have been associated with the olfactory system. TOMM40, the gene in linkage disequilibrium with APOE4, emerged as being associated with tau neurodegeneration. APOE4 is the undisputed genetic association for AD pathology and the models for this study have not incorporated APOE genotype. This raises the question whether the OID and tau association is driven by the latent association with APOE4. All statistical models should be rerun with APOE4 genotype in the model.

Thanks to this important reviewer's comment, we have rerun all the analysis controlling for APOE4 status as suggested. Supplementary Figure 1 shows the results controlling for APOE4 status. We obtained similar findings, but as this is relevant information, we have placed it in the main text and in supplementary material.

Supplementary Fig 1

UPSIT smell test & tau at baseline

Supplementary Fig. 1 | Smell identification test association with tau accumulation adjusting for atrophy, amyloid, and APOE4 status.

Reviewer #2

Overall comments

This interesting and ambitious study sought to examine, in a population of older persons, associations between differences in olfactory function, as measured by a well-validated smell test, and patterns of brain neurodegeneration and connections among brain structures, as well as the spreading of tau within selected olfaction-related brain regions. The authors suggest that tau accumulation within the medial temporal region is associated with early stages of Alzheimer's disease, resulting in smell loss observed at that time. Co-expression of odor- and tau-related genes is noted. The general conclusion is that tau emerges from the limbic system and only secondarily involves olfactory structures that would be implicated if an exogenous agent entered the olfactory pathways from the nose. The work is largely novel and appears to be well performed.

We sincerely appreciate the reviewer's enthusiasm and thoughtful critique of this work. In the following pages, we respond to each point with clarifications to address all the reviewer's comments.

Specific comments

Abstract: The abstract could benefit with more detail – numbers of subjects, age range, procedures, etc. Presently it does not provide even minimal description of what was actually done.

We thank the reviewer and agree that the abstract would benefit from the inclusion of more details. We updated the abstract accordingly:

The direct access of olfactory afferents to memory-related cortical systems has inspired theories about the role of the olfactory pathways in the development of cortical neurodegeneration in Alzheimer's disease (AD). In this study, we used longitudinal flortaucipir and PiB PET, diffusion MRI, olfaction identification measures of 89 cognitively normal older adults (73.82±8.44 yo; 56% females), and a transcriptomic data atlas to investigate the spatiotemporal spreading and genetic vulnerabilities of AD-related pathology aggregates in the olfactory system. We find that odor identification deficits predominantly associate with tau accumulation in key areas of the olfactory pathway, with a particularly strong predictive power for tau longitudinal progression. We observe that tau spreads from the medial temporal lobe structures toward the olfactory system -and not the reverse-. Moreover, we describe new observations regarding a genetic background of odor perception-related genes that might confer vulnerability to tau accumulation along the olfactory system.

Introduction, p. 4, lines 51-7: While the olfactory system is unique in terms of largely bypassing the thalamus before reaching cortical regions, the other major sensory systems also involve the entorhinal cortex and the locus coeruleus. Hence olfaction is not unique in this regard. This section should be reworded accordingly.

Thanks to the reviewer's comment, we have modified the introduction accordingly.

p. 4, line 19. Perhaps want to also consider other xenobiotics and reword something like, "...a link with viral and other xenobiotic impacts on the olfactory pathways." I am surprised that none of the extensive work of Calderón-Garcidueñas et al. is considered or even mentioned in the introduction or discussion (e.g Calderón-Garcidueñas et al., Urban air pollution: Influences on olfactory function and pathology in exposed children and young adults. *Experimental and Toxicological Pathology*, 62:91-102, 2010). See also O'Piela et al., Particulate matter and Alzheimer's disease: an intimate connection. *Trends in Molecular Medicine* 2022 Sep;28(9):770-780. Wouldn't these types of studies add potential fuel to the concept of entrance of agents into the olfactory system relative to AD? How do Calderón-Garcidueñas et al.'s results fit with the data and the general theory subscribed to in the present study? Does tau have to be the final arbiter?

Thanks to the reviewer's comment, we have included this point in the introduction and discussion to help fuel the manuscript's narrative. We have also incorporated the suggested citations [1,2].

We do not think tau in the olfactory system is directly associated with infections or other xenobiotics. Instead, we believe that infections and xenobiotics activate the immune system and microglia/astrocytes and that a prolonged activation leads to neuroinflammation and aggregation of amyloid [1,3]. We believe that the increased amyloid and inflammation related to xenobiotics accelerates tau progression from medial temporal regions to the olfactory system and leads to neurodegeneration. While Calderón-Garcidueñas reports increased p-tau and A β in children and young adults with high exposure to PM [4], this might be due to accelerated neurodegeneration led by increased A β and neuroinflammation. This is partly supported by other studies finding associations between PM and tau only in amyloid-positive cases [5] or finding associations with A β but not with tau [6]. To further explore this theory, we performed a new analysis incorporating olfactory measures and amyloid and tau PET.

We computed the association of every voxel tau with olfaction and every voxel amyloid with olfaction adjusting for age, sex, and smoking history (N=155; Figure 3). We applied principal component analysis to reduce the amount of information and study the component maximizing the explained variance between the association of olfactory deficits with voxel-level tau and amyloid accumulation. Using this strategy, we obtained an OID biomarker associated with tau accumulation in medial temporal and olfactory regions and widespread amyloid accumulation (Supplementary Figure 2). While both amyloid and tau could be predicted with odorant information, tau showed stronger associations and survived even after controlling for amyloid; amyloid did not survive when controlling for tau (Figure 3). This suggests that tau is strongly associated with olfactory dysfunction even when amyloid is present.

Supplementary Figure 2

Regions of increased tau and amyloid accumulation explaining most of the variance associated with identification of 40 individual odorants

Supplementary Fig. 2 | Voxel-level association between tau and amyloid PET with a linear combination of individual UPSIT items that maximized voxel-level tau and amyloid prediction.

[1]. O’Piela, D. R., Durisek, G. R., Escobar, Y.-N. H., Mackos, A. R. & Wold, L. E. Particulate matter and Alzheimer’s disease: an intimate connection. *Trends Mol Med* 28, 770–780 (2022).

[2]. Calderón-Garcidueñas, L. *et al.* Urban air pollution: Influences on olfactory function and pathology in exposed children and young adults. *Exp Toxicol Pathol* 62, 91–102 (2010).

[3]. Iaccarino, L. *et al.* Association Between Ambient Air Pollution and Amyloid Positron Emission Tomography Positivity in Older Adults With Cognitive Impairment. *Jama Neurol* 78, 197–207 (2021).

[4]. Calderón-Garcidueñas, L. *et al.* Interactive and additive influences of Gender, BMI and Apolipoprotein 4 on cognition in children chronically exposed to high concentrations of PM_{2.5} and ozone. APOE 4 females are at highest risk in Mexico City. *Environ Res* 150, 411–422 (2016).

[5]. Alemany, S. *et al.* Associations between air pollution and biomarkers of Alzheimer’s disease in cognitively unimpaired individuals. *Environ Int* 157, 106864 (2021).

[6]. Lee, H. W., Kang, S.-C., Kim, S.-Y., Cho, Y.-J. & Hwang, S. Long-term Exposure to PM₁₀ Increases Lung Cancer Risks: A Cohort Analysis. *Cancer Res Treat* 54, 1030–1037 (2022).

Results, p. 7, lines 15+; p. 9, lines 8-15. Attempts to identify specific UPSIT items involved in a number of neurodegenerative diseases have been contradictory and have shed little light on the association between smell and such diseases. It is not clear that seeking to identify specific UPSIT items relative to tau or Braak stages has much scientific value. Indeed, earlier studies have lacked uniformity of findings across UPSIT items (e.g., for Parkinson’s disease). This reflects, in part, the fact that the different UPSIT stimuli are not iso-intense and many are individually made up of numerous chemical compounds. Hence, names like “rose”, “dill pickle” etc. would not appear to have much operational scientific value. It appears that the UPSIT stimuli that were most correctly identified in this study were the more intense odors. This problem is complicated by a number of other factors. For example, receptor types of unknown ratios are present in the epithelium, some of which are responsive to more than one defined chemical, i.e., are so-called generalists. Cells expressing the same receptor type may have different tuning curves, and the

dynamic ranges of various chemicals vary. Hence, my suggestion would be to either address these issues in the discussion or omit these elements from the paper.

We thank the reviewer for raising this important point. Following the reviewer's suggestion, we have moved Figure 3 to supplementary material (Supplementary Figure 4). Moreover, we performed a new analysis and added a new figure to the manuscript (Figure 3), searching for linear combinations of the 40 odorants that maximize the association with voxel-level tau and amyloid accumulation. Instead of focusing on specific odorant associations, we generated cross-sectional and longitudinal biomarkers that best predict voxel-level tau and amyloid accumulation. The new Figure 3 displays different linear combinations of all the odorants – biomarkers – and the expected tau accumulation. Additionally, we added Supplementary Figure 3, showing the Jaccard similarity index between all odors (N=418) to fully described these results. Additionally, as suggested by the reviewer, we decided to include all these limitations in the discussion, involving the combinatory/grouping effects in more detail in the new version of the manuscript.

Fig. 3 | Olfactory biomarkers for tau accumulation and longitudinal increase. (A) Linear combination of 40 odorants (olfactory biomarkers) maximizing explained variance of cross-sectional voxel-wise amyloid and tau accumulation prediction (Cross1; N=155). In different brain circuits, three other linear combinations predict longitudinal tau increase in 2.5 years (Long1-3; N=89). (B) Z-stat of the association of Cross1 olfactory biomarker and tau accumulation in the brain. Results survived when adjusting for amyloid. See supplementary figure 2 for association with amyloid. (C) Three components - different odorant linear combinations – predict longitudinal tau increase in the regions projected on the surface. Only areas surviving multiple comparisons when computing the association between tau accumulation and olfactory biomarkers are displayed. All the previous associations were adjusted for age, sex, and smoking history.

Supplementary Fig 3

Supplementary Fig. 3 | Similarity (Jaccard index) between the UPSIT items identification ability in N=418.

Results, p. 8, lines 12+. Why was smell only performed at baseline? While it is interesting to determine whether the baseline UPSIT score could predict tau progression over time, it would seem similarly interesting to determine whether the smell test score changed in the same manner as the change in tau. Perhaps this could be addressed in the discussion section.

We thank the reviewer for the opportunity to clarify this point. The HABS study acquired UPSIT once in year 3, concurrently with tau and amyloid PET imaging. Unfortunately, we do not have the olfactory measure performed with the longitudinal tau imaging. We agree with the reviewer that a follow-up UPSIT would have been optimal to explore further changes in smell perception. We have added this point as a limitation in the discussion.

Results, p. 9-11. It appears that a number of assumptions have to be made in the various analyses in order to come to the conclusions. Mentioning these assumptions may be helpful to readers who lack detailed knowledge of these types of analyses. For example,

can one truly infer that by measuring the amount of tau at two points in time in different brain regions that tau is spreading between them? According to the title, Tau is propagating through olfactory brain circuits. Are there alternative explanations, such as development of differential sensitivity of brain regions to tau formation over time?

Following the reviewer's suggestions, we have included extensive methodological details and two references in the *Directional Graph Theory Regression of Tau Accumulation* section to better explain the results' rationale and interpretation. For instance, we have clarified how the tau accumulation in one region is associated with voxel-level tau increase 2 years later. We performed this by looking at how tau covariates longitudinally. To ensure that we minimize cross-sectional associations and just focus on longitudinal tau increase, we controlled the longitudinal connectivity by the tau accumulation at the baseline of the evaluated longitudinal region. Thus, differential sensitivity between brain regions does not affect the results, as we look at the data's covariance and not the signal's amplitude. On the other hand, we agree that the noise generated by off-target binding could have weakened the associations. However, we will not likely get overestimated connectivity values as we control for baseline tau removing the noise in those specific regions, minimizing the spurious correlations of the off-target binding noise.

We agree that assuming the amount of tau between two longitudinal points might not represent spreading between them. If we have a brain region spreading tau to the other two, we will find a spreading measure between these nodes (indirect connection). To address this problem, we additionally applied conditional independence testing to remove these indirect connections from our network, using the skeleton identification step of the PC-algorithm. We extended the methodological section and discussion to elaborate on these assumptions.

It would be helpful to have a paragraph at the end of the discussion outlining the strengths and limitations of various elements of the study. It is my understanding, for example, that there are limitations of using gene expression data in attempts to identify functional associations and perhaps this can be articulated in this paragraph.

We thank the reviewer for pointing out this critical point. Thanks to the reviewer's suggestion, we included such a paragraph in the discussion to address the limitations related to sample size, availability of variables, off-target binding issues, assumptions on tau spreading, and constraints of using gene expression data, among others.

Reviewer #3

Overall comments

Olfactory deficits are known to be related to tau pathology in the brain, particularly in Alzheimer's disease and Lewy body disease. The authors examine associations between individual items on the UPSIT and Flortaucipir tau PET imaging. While this is an area definitely worth investigating, the results do not answer the main questions in a clear and logical manner.

We thank the reviewer for taking the time to evaluate the merits and limitations of this paper rigorously. We have worked extensively to improve the manuscript's clarity. In the following pages, we have addressed the reviewer's comments point-by-point.

Specific comments

1. There were 418 cognitively normal individuals studied, 155 of them got tau imaging at baseline and 89 at follow-up an average of 2.5 years later. There is no clear explanation of which of these samples were examined in which analysis in the text nor in the figure legends. Baseline cross-sectional analyses are not distinguished from longitudinal analyses in drawing the conclusions that are made. This is important because these were cognitively normal individuals and not much change in olfaction or tau imaging would be expected over 2.5 years.

Thanks to the reviewer's comment, we have incorporated the sample size used in each analysis in the text and Figure legends. Moreover, we would like to note that Figure 2 shows the difference between the cross-sectional and longitudinal association of UPSIT and tau of the same participants (N=89; please see on the right the overlap between them; blue=cross-sectional; green=longitudinal). As noted by the reviewer, the association after 2.5 years is very similar, but it extends to the left hemisphere and prefrontal cortex. Additionally, a new Figure 3 has been added in the manuscript to

show the linear combinations of odorants that maximize the prediction of voxel-level tau and amyloid increase at baseline and longitudinal follow-up. One of the principal components is associated with increased tau in medial temporal and olfactory regions. Still, the other two components are associated with tau increase in later Braak stages. We believe this information provides new evidence obtained from the longitudinal assessment.

2. The claim that olfaction is related to tau PET and not amyloid PET indices can be ascertained by a simple analysis involving these three measures, but this is not directly described.

We thank the reviewer for raising this important point. Thanks to this suggestion, we have added a new figure (Figure 3) that addresses the point in the revised version of the manuscript.

In Figure 3, we computed the association of every voxel of tau with olfaction, as well as every voxel of amyloid with olfaction (N=155). These associations were adjusted for age, sex, and smoking history. We applied a principal component analysis to study the association between tau, amyloid, and olfaction. We obtained all components that maximize the explained variance between olfactory scores and voxel-level tau or amyloid accumulation. While both tau and amyloid cortical accumulation predicted odorant information, tau showed stronger associations. Furthermore, the association with tau survived even after controlling for amyloid. On the contrary, amyloid associations did not survive when controlling for tau (please see more details in the results section). Of note, the new Supplementary Figure 1 shows the association between UPSIT and tau accumulation after controlling the effect of amyloid. In each voxel, the impact of amyloid was removed from the tau-UPSIT associations, attaining similar results as Figure 2.

Fig. 3 | Olfactory biomarkers for tau accumulation and longitudinal increase. (A) Linear combination of 40 odorants (olfactory biomarkers) maximizing explained variance of cross-sectional voxel-wise amyloid and tau accumulation prediction (Cross1; N=155). In different brain circuits, three other linear combinations predict longitudinal tau increase in 2.5 years (Long1-3; N=89). (B) Z-stat of the association of Cross1 olfactory biomarker and tau accumulation in the brain. Results survived when adjusting for amyloid. See supplementary figure 2 for association with amyloid. (C) Three components - different odorant linear combinations – predict longitudinal tau increase in the regions projected on the surface. Only areas surviving multiple comparisons when computing the association between tau accumulation and olfactory biomarkers are displayed. All the previous associations were adjusted for age, sex, and smoking history.

Supplementary Fig 1

UPSIT smell test & tau at baseline

Supplementary Fig. 1 | Smell identification test association with tau accumulation adjusting for atrophy, amyloid, and APOE4 status.

3. Flortaucipir shows off-target binding with PET. There is no discussion of this issue or the limitations of the study, which include only 21% of the original sample getting follow-up tau PET scans. There is no description of how the missing data were handled.

We appreciate the reviewer's comment and recognize this omission of detail. Following these suggestions, we have included a new paragraph with the study's limitations, including the off-target binding of flortaucipir and smaller sample size for longitudinal data analysis, in a conclusion section. For all the analyses that included longitudinal data, we limited the study to 89 participants with both time points and did not use the other 66 or estimate/interpolate data.

4. The UPSIT total score is the conventional measure to analyze for this test. The correlation between the total UPSIT score and the tau binding (total and ROIs) is not presented, suggesting that it was not robust enough to describe. Instead, the correlations for each of 40 UPSIT odorants with tau imaging parameters is the focus of the paper.

The reviewer is correct; the total UPSIT score is the conventional measure. We used the total UPSIT score in the core figure of the manuscript (Figure 2). Figure 2 shows the association between the total UPSIT score and voxel-level tau binding. Our work focused on investigating the high-resolution cortical mapping of the total UPSIT score using voxel-level statistical analysis. We will be delighted to provide global or regional values of cortical areas, as they can be easily derived from high-resolution maps. Additionally, Figure 4a shows the association of the total UPSIT score with diffusion parameters along tracts associated with the olfactory system.

5. Did baseline total UPSIT correlate with baseline tau imaging parameters (total binding, ROI binding) and did it predict change in tau imaging parameters? Did the change in total UPSIT total score track directly with the change in tau measures over the 2.5 years of follow-up? These results are essential (but missing) to address the questions posed by the authors. The associations described with sophisticated image analysis are difficult to

interpret in the absence of a description of the correlation coefficients among the main measures of interest.

Following the reviewer's suggestion, we added a new Supplementary Table 5 showing the tau-UPSIT total score associations in medial temporal and olfactory regions, both cross-sectional and longitudinal.

	Cross-sectional		Longitudinal	
	Left	Right	Left	Right
Parahippocampus	T=1.46; p=0.15	T=1.92; p=0.06	T=2.31; p=0.02	T=2.90; p<0.01
Entorhinal	T=1.37; p=0.17	T=2.63; p=0.01	T=2.10; p=0.03	T=2.78; p<0.01
Amygdala	T=1.26; p=0.21	T=2.39; p=0.02	T=1.44; p=0.15	T=2.99; p<0.01
Hippocampus	T=2.19; p=0.03	T=2.34; p=0.02	T=2.73; p<0.01	T=2.92; p<0.01
Piriform	T=1.97; p=0.05	T=2.70; p<0.01	T=1.72; p=0.09	T=3.00; p<0.01
TUB	T=0.27; p=0.78	T=1.50; p=0.14	T=2.73; p<0.01	T=2.49; p=0.01
AON	T=2.50; p=0.01	T=2.46; p=0.01	T=2.54; p=0.01	T=3.10; p<0.01

6. Which covariates were used? Do the findings remain after covarying for age?

All the results were adjusted for age, sex, and smoking history. Additionally, for association with PACC5, we adjusted for years of education. We also performed statistical analyses adjusting for atrophy, amyloid, or APOE4 carrier status in Supplementary Figure 1. Thanks to the reviewer's suggestion, we have extended the explanation in the corresponding sections of the text.

7. A quadratic rather than a linear function was used. No explanation is provided for this choice.

We thank the reviewer for this comment. We understand the reviewer's comment refers to our manuscript's Figure 1a (UPSIT vs. age). The scatterplot displayed a curvilinear rather than a linear pattern in this analysis. Therefore, we applied curvilinear models and observed that the quadratic approach fit better (R values). We think this is a standard approach in curvilinear modeling.

8. A novel graph theory approach was used. There is no description of what this involves and whether any validation has been published.

Thanks to the reviewer's comment, we have included the missing references of our validated graph theory method:

"In contrast to conventional analysis approaches in PET imaging examining regional binding differences, we used a recently developed graph theory approach to identify the directionality of tau spreading using longitudinal tau connectivity[1]. Tau connectivity measures how a tau signal in a given region relates to tau signal in another region[2], locally or distantly located. Further extending this approach to longitudinal data, we can study how tau at baseline is associated with longitudinal tau increase in any brain voxel. To ensure we minimize cross-sectional associations and only look to associations with a longitudinal increase in tau, we controlled the longitudinal connectivity by the tau accumulation at the baseline of the evaluated longitudinal region. Compared with previous approaches, this correction assures the connectivity reflects longitudinal spreading and not tau accumulation covariance."

[1]. Sepulcre, J. *et al.* Neurogenetic contributions to amyloid beta and tau spreading in the human cortex. *Nature Medicine* 24, 1910–1918 (2018).

[2]. Sala, A. *et al.* Brain connectomics: time for a molecular imaging perspective? *Trends Cogn Sci* (2023) doi:10.1016/j.tics.2022.11.015.

9. Overall, the analyses are based on a large number of assumptions without the presentation of basic statistics about the associations between the variables of interest and how they change over time.

Following the reviewer's comments #9 and #5, we have added a new Supplementary Table showing basic statistics of the core finding of the manuscript: the tau-UPSIT total score associations in medial temporal and olfactory regions, both cross-sectional and longitudinal levels.

10. Genes derived from a dataset in the literature from 6 individuals are analyzed. These analyses do not seem to be related to the main point of the paper regarding the associations between olfaction and tau imaging.

We appreciate the reviewer's comment about the genetic findings. In this project, we have two main aims: 1) to characterize tau spreading patterns related to olfaction impairment and 2) to provide insights into potential propagation axis from olfactory-to-central regions or from central-to-olfactory regions. Due to our results supporting the notion that central-to-olfactory propagation is the likely mechanism, we were driven to investigate whether there are biological factors of an intrinsic vulnerability of the olfactory systems to develop tau pathology. Thus, we believe our neuroimaging-genetics approach is a relevant tool for identifying molecular mechanisms related to the spreading of the disease [3] in the olfactory system. For instance, it is intriguing to see that besides the olfactory regions, olfactory genes are highly expressed in areas where tau is accumulated in AD, suggesting that a common mechanism could be intersecting. Please see the following Figure:

1]. Sepulcre, J. *et al.* Neurogenetic contributions to amyloid beta and tau spreading in the human cortex. *Nature Medicine* 24, 1910–1918 (2018).

[2]. Sala, A. *et al.* Brain connectomics: time for a molecular imaging perspective? *Trends Cogn Sci* (2023) doi:10.1016/j.tics.2022.11.015.

[3] Diez, I. & Sepulcre, J. Unveiling the neuroimaging-genetic intersections in the human brain. *Curr Opin Neurol* Publish Ahead of Print, (2021).

Reviewer #1 (Remarks to the Author):

Regarding the importance of the paper it is a matter of opinion, I still don't see any transformative observation that will change how we think about the relationship between olfactory deficit, aging and neurodegeneration.

APOE4 incorporation to the analysis is insufficient at present, only used as a post hoc adjustment in one model (Supp. Fig. 1). It needs to be done properly to eliminate the latent association driving the results, that APOE4 is associated with neurodegeneration. It is even more important as the authors themselves identified TOMM40 as the most robust genetic association with olfactory deficit and all aspects of neurodegeneration and TOMM40 is in linkage disequilibrium with APOE4. At this point all findings could be driven by APOE4.

P6 line 19-20: please revise model incorporating all relevant covariates age, sex, smoking history and APOE4

P7 line 5-6: please revise model incorporating all relevant covariates controlling for age, gender, smoking history, years of education and APOE4

P7 line 15-16: please revise model incorporating all relevant covariates controlling for age, sex, smoking history and APOE4

P9 line 17-19: please revise model incorporating all relevant covariates age, sex, APOE4

P10 line 7-8: please revise model incorporating all relevant covariates age, sex, APOE4 "For each pair of

8 olfactory, medial temporal, and brainstem regions, we computed the linear regression"

P11 line 10-13: genetic models should also incorporate covariates; please revise the genetic model with covariates age, sex and APOE4.

Reviewer #2 (Remarks to the Author):

Because of its complexity, I find sections of the paper difficult to understand. I have only two general comments.

First, in sections discussing "odors", it would seem better to discuss "odor items" since this is what is being assessed in regards to the UPSIT (e.g., p. 6, line 22; p. 7, line 2). This should be adjusted throughout the paper.

Second, I found Figure 3A difficult to understand. The nomenclature seems problematic (e.g., what does Long 1, 2 and 3 really mean? Can these be called something else?). While these are linear combinations of odorants, the whole diagram is not intuitive and is difficult to follow. The caption of the figure does not help much in this regard. This is perhaps an artifact of not having the information on p. 23 come before the results but this somehow needs to be better initially explained.

Reviewer #3 (Remarks to the Author):

The authors have responded in detail to the reviews. Some concerns remain.

Causal inferences about the direction of propagation of tau pathology is derived from an olfactory assessment at a single time-point and tau PET imaging at two time-points. This major limitation requires the findings to be considered preliminary and they need to be replicated with serial assessment of both olfaction and tau PET imaging at multiple time-points. The language used in the Abstract is misleading, e.g., "we used longitudinal flortaucipir and PiB PET, diffusion MRI, olfaction identification measures of 89 cognitively normal older adults (73.82 ± 8.44 yo; 56% females)," leaves the reader with the impression that olfaction was assessed longitudinally, which it was not.

The limitations described by the reviewers have not been adequately addressed in the discussion

of limitations.

The number of analyses was large and is now even larger with several supplementary tables and figures, some of which are not directly related to the main point of the paper.

Genetic analysis and conclusions from a sample of 6 individuals is not justified.

Reviewer #1

Regarding the importance of the paper it is a matter of opinion, I still don't see any transformative observation that will change how we think about the relationship between olfactory deficit, aging and neurodegeneration.

We thank the reviewer for taking the time to review the manuscript again. This manuscript presents novel cortical mapping and tau-spreading observations that are unprecedented in the literature. Particularly, these observations include:

- The association of olfactory identification with longitudinal brain spatiotemporal tau spreading.
- The study of tau progression in different pathways connecting to olfactory structures in order to disentangle olfactory-specific spreading patterns.
- The proposal of odor items as biomarkers associated with tau progression stages.
- The cortical mapping of transcriptomic data providing mechanistic insights into the vulnerability of the olfactory system to accumulate tau.

We believe these findings significantly advance our understanding of the olfactory changes in aging and neurodegeneration.

We will be delighted to further clarify the paper's novelty, if necessary, based on further specific literature information provided by the reviewer.

APOE4 incorporation to the analysis is insufficient at present, only used as a post hoc adjustment in one model (Supp. Fig. 1). It needs to be done properly to eliminate the latent association driving the results, that APOE4 is associated with neurodegeneration. It is even more important as the authors themselves identified TOMM40 as the most robust genetic association with olfactory deficit and all aspects of neurodegeneration and TOMM40 is in linkage disequilibrium with APOE4. At this point all findings could be driven by APOE4.

- **P6 line 19-20: please revise model incorporating all relevant covariates age, sex, smoking history and APOE4**
- **P7 line 5-6: please revise model incorporating all relevant covariates controlling for age, gender, smoking history, years of education and APOE4**
- **P7 line 15-16: please revise model incorporating all relevant covariates controlling for age, sex, smoking history and APOE4**
- **P9 line 17-19: please revise model incorporating all relevant covariates age, sex, APOE4**
- **P10 line 7-8: please revise model incorporating all relevant covariates age, sex, APOE4 “For each pair of olfactory, medial temporal, and brainstem regions, we computed the linear regression”**
- **P11 line 10-13: genetic models should also incorporate covariates; please revise the genetic model with covariates age, sex and APOE4.**

Following the reviewer's suggestions, we have replicated all the study findings correcting for APOE4 status. This is now included in Supplementary Figure 1 (see below). APOE4 status does not substantially affect the associations of olfactory identification measures with tau spreading.

Replication adjusting for APOE4 status

(A) Olfactory system dysfunction in healthy aging and its association to neurodegeneration

(B) Smell identification test association with tau accumulation and spreading

(C) Longitudinal tau progression

Reviewer #2

Because of its complexity, I find sections of the paper difficult to understand. I have only two general comments.

We would like to thank the reviewer for the valuable comments. We have rewritten several sections of the manuscript to improve the clarity and readability of our findings. We believe that these revisions have strengthened the overall narrative.

First, in sections discussing "odors", it would seem better to discuss "odor items" since this is what is being assessed in regards to the UPSIT (e.g., p. 6, line 22; p. 7, line 2). This should be adjusted throughout the paper.

Thank you for pointing this out. We have updated all instances of "odors" to "odor items."

Second, I found Figure 3A difficult to understand. The nomenclature seems problematic (e.g., what does Long 1, 2 and 3 really mean? Can these be called something else?). While these are linear combinations of odorants, the whole diagram is not intuitive and is difficult to follow. The caption of the figure does not help much in this regard. This is perhaps an artifact of not having the information on p. 23 come before the results but this somehow needs to be better initially explained.

Thanks to the reviewer's comment, we have revised this section to provide a more concise and understandable explanation. Specifically, we have made the following changes:

- We have included a detailed explanation of the analysis in the results section. This will allow readers to understand directly without revisiting the methods section.
- We have rearranged Figure 3 to make it easier to follow visually.
- We have rewritten the figure legend to add additional clarity.

Reviewer #3

The authors have responded in detail to the reviews. Some concerns remain.

We would like to thank the reviewer for the thoughtful comments. We have made several changes to the manuscript to include the required clarifications and address the raised concerns.

Causal inferences about the direction of propagation of tau pathology is derived from an olfactory assessment at a single time-point and tau PET imaging at two time-points. This major limitation requires the findings to be considered preliminary, and they need to be replicated with serial assessment of both olfaction and tau PET imaging at multiple time-points. The language used in the Abstract is misleading, e.g., "we used longitudinal flortaucipir and PiB PET, diffusion MRI, olfaction identification measures of 89 cognitively normal older adults (73.82±8.44 yo; 56% females)," leaves the reader with the impression that olfaction was assessed longitudinally, which it was not.

We updated the abstract to make clear we used baseline olfactory measures with longitudinal PET in this study.

The limitations described by the reviewers have not been adequately addressed in the discussion of limitations.

We have added a sentence to the limitations section of the manuscript to state the preliminary nature of the results and the need for further validation using longitudinal olfactory and PET measures.

The number of analyses was large and is now even larger with several supplementary tables and figures, some of which are not directly related to the main point of the paper.

We agree with the reviewer on this point. However, it is essential to remember that most of the new analyses have been incorporated to solve the other reviewers' suggestions.

Genetic analysis and conclusions from a sample of 6 individuals is not justified.

Our genetic analyses support that specific olfactory biological vulnerabilities exist for tau accumulation in the aging brain. The first analysis used GWAS catalog associations, which provide genetic information from thousands of participants. This strategy offers strong evidence of existing associations between mutations that increase tau accumulation and changes in the expression of olfactory perception genes. Then, the AHBA data -based on the transcriptomic data of 6 individuals- were used to provide the spatial domain needed to find the brain regions most likely impacted. Following the reviewer's suggestion, we noted in the text that AHBA has several limitations, including the sample size.

Reviewer #1 (Remarks to the Author):

Please indicate p-value after statements in the text "remained significant after correction for APOE4".

Please add full model p-values to Supp. Table 5.

Please revise methods section to describe the full statistical model that incorporated all necessary covariates:

P6 line 19-20: please revise model incorporating all relevant covariates age, sex, smoking history and APOE4

P7 line 5-6: please revise model incorporating all relevant covariates controlling for age, gender, smoking history, years of education and APOE4

P7 line 15-16: please revise model incorporating all relevant covariates controlling for age, sex, smoking history and APOE4

P9 line 17-19: please revise model incorporating all relevant covariates age, sex, APOE4

P10 line 7-8: please revise model incorporating all relevant covariates age, sex, APOE4 "For each pair of

8 olfactory, medial temporal, and brainstem regions, we computed the linear regression"

P11 line 10-13: genetic models should also incorporate covariates; please revise the genetic model with covariates age, sex and APOE4.

Please add to Supplementary Table 5 the full model p-values.

Reviewer #1

We would like to thank the reviewer for the comments and the opportunity to improve our manuscript. We have updated the text according to their suggestions (red font color in the main text).

Please indicate p-value after statements in the text “remained significant after correction for APOE4”.

We thank the reviewer for the comment. We have included the statistics and p-values for ROI-based analyses after correcting for APOE4. For the voxel-level neuroimaging analyses, the figures show all significant voxels below a p-value < 0.05 , corrected for multiple comparisons. Therefore, in these cases, we report the entire final map as an image, in which each voxel has a corresponding p-value.

Please add full model p-values to Supp. Table 5.

Following the reviewer’s suggestions, we have updated Table 5, which now includes the model p-values adjusting for APOE4.

Please revise methods section to describe the full statistical model that incorporated all necessary covariates:

- **P6 line 19-20: please revise model incorporating all relevant covariates age, sex, smoking history and APOE4**
- **P7 line 5-6: please revise model incorporating all relevant covariates controlling for age, gender, smoking history, years of education and APOE4**
- **P7 line 15-16: please revise model incorporating all relevant covariates controlling for age, sex, smoking history and APOE4**
- **P9 line 17-19: please revise model incorporating all relevant covariates age, sex, APOE4**
- **P10 line 7-8: please revise model incorporating all relevant covariates age, sex, APOE4 “For each pair of 8 olfactory, medial temporal, and brainstem regions, we computed the linear regression”**
- **P11 line 10-13: genetic models should also incorporate covariates; please revise the genetic model with covariates age, sex and APOE4.**

Following the reviewer’s suggestions, we have revisited and modified the main text, and the manuscript now includes the full statistical models with all the covariates, particularly clarifying the inclusion of APOE4. As for the genetic data, we have used the summary statistics of already published GWAS studies in the literature. These GWAS statistics already incorporate the correction for major genetic confounds, including APOE4.